# The *Wolbachia* cytoplasmic incompatibility enzyme CidB targets nuclear import and protamine-histone exchange factors

John Frederick Beckmann[1]*, Gagan Deep Sharma[1], Luis Mendez[1], Hongli Chen[2], Mark Hochstrasser[2,3]*

[1]Department of Entomology and Plant Pathology, Auburn University, Auburn, United States; [2]Department of Molecular Biophysics and Biochemistry, Yale University, New Haven, United States; [3]Department of Molecular, Cellular, and Developmental Biology, Yale University, New Haven, United States

**Abstract** Intracellular *Wolbachia* bacteria manipulate arthropod reproduction to promote their own inheritance. The most prevalent mechanism, cytoplasmic incompatibility (CI), traces to a *Wolbachia* deubiquitylase, CidB, and CidA. CidB has properties of a toxin, while CidA binds CidB and rescues embryonic viability. CidB is also toxic to yeast where we identified both host effects and high-copy suppressors of toxicity. The strongest suppressor was karyopherin-α, a nuclear-import receptor; this required nuclear localization-signal binding. A protein-interaction screen of *Drosophila* extracts using a substrate-trapping catalytic mutant, CidB*, also identified karyopherin-α; the P32 protamine-histone exchange factor bound as well. When CidB* bound CidA, these host protein interactions disappeared. These associations would place CidB at the zygotic male pronucleus where CI defects first manifest. Overexpression of karyopherin-α, P32, or CidA in female flies suppressed CI. We propose that CidB targets nuclear-protein import and protamine-histone exchange and that CidA rescues embryos by restricting CidB access to its targets.

*For correspondence:
beckmann@auburn.edu (JFB);
mark.hochstrasser@yale.edu (MH)

**Competing interests:** The authors declare that no competing interests exist.

## Introduction

*Wolbachia* are obligate intracellular bacteria infecting arthropods and filarial nematodes (*Werren et al., 2008*). They promote their maternal transmission by reproductive manipulations, most commonly cytoplasmic incompatibility (CI) (*Beckmann et al., 2019a*; *Beckmann and Fallon, 2013*; *Beckmann et al., 2017*; *Chen et al., 2019*). CI causes zygotic lethality when infected males mate with uninfected females (*Ferree and Sullivan, 2006*; *Presgraves, 2000*). If females have matching infections, embryo viability is normal (*Poinsot et al., 2003*). It is thus a gene drive mechanism that selects for infected females.

CI was first studied in the mosquito *Culex pipiens*, which harbors a *Wolbachia* endosymbiont correspondingly named *w*Pip (*Laven, 1953*; *Laven, 1967a*). Later, Yen and Barr implicated *Wolbachia* as the CI inducer (*Yen and Barr, 1971*; *Yen and Barr, 1973*). Several related models have been proposed for the CI mechanism (*Beckmann et al., 2019a*; *Werren, 1997*; *Shropshire et al., 2018*). CI is being applied in mosquito control to sterilize mosquitoes (*Bushland et al., 1955*; *Laven, 1967b*; *Mains et al., 2016*; *Zheng et al., 2019*) and as a population replacement tool harnessing *Wolbachia's* ability to inhibit infectious agents such as dengue and Zika viruses (*Turelli and Hoffmann, 1991*; *Schmidt et al., 2017*; *Walker et al., 2011*).

In a cross between compatible male and female insects, zygotes follow well described developmental pathways (*Loppin et al., 2015*; *Serbus et al., 2008*). An early step is nuclear envelope

breakdown (NEB) of the sperm-derived male pronucleus. The small, highly basic protamine proteins used to package paternal DNA at high density are stripped from the DNA, (*Balhorn, 2007*; *Rathke et al., 2014*; *Tirmarche et al., 2014*; *Loppin et al., 2015*; *Tirmarche et al., 2016*) and nucleosomes are then assembled with maternal histones (*Loppin et al., 2015*; *Liu et al., 1997*). The protamine-histone transition utilizes specific histone chaperones such as P32 and Nap1 (*Emelyanov et al., 2014*; *Emelyanov and Fyodorov, 2016*). Subsequently, male and female pronuclei come together (but do not fuse) and undergo DNA replication. In the first zygotic mitosis, the two sets of chromosomes condense, align on the metaphase plate, separate in anaphase in parallel and then finally intermingle (*Tram et al., 2003*).

In CI zygotes, the earliest detected abnormality is impaired maternal H3.3 histone deposition onto the paternal DNA following protamine removal (*Landmann et al., 2009*). Paternal pronuclear NEB is delayed and activity of the cell-cycle kinase CDK1, which normally drives the metaphase-to-anaphase transition, is inhibited in the male pronucleus (*Tram and Sullivan, 2002*). Condensation of the paternal chromosomes is delayed or impaired, often leading to chromosome shearing and bridging during anaphase (*Callaini et al., 1997*; *Reed and Werren, 1995*; *Ryan and Saul, 1968*). This is fatal in diploid insects.

Similar CI cytology has been documented in diverse insects (*Tram et al., 2003*). Furthermore, artificial transfer of heterologous *Wolbachia* strains into different insect species usually still causes CI (*Bian et al., 2013*; *Boyle et al., 1993*; *Ye et al., 2015*). The phenotypic consistency across species suggests that *Wolbachia*-induced CI targets conserved cellular machinery required for cell and nuclear division (*Landmann et al., 2009*; *Callaini et al., 1997*; *Reed and Werren, 1995*; *Callaini et al., 1996*). CI might directly disrupt the protamine-histone exchange (*Landmann et al., 2009*); other extra-nuclear sperm factors have been ruled out as targets (*Presgraves, 2000*). From the results in the current study, we propose that key CI targets include nuclear transport factors (karyopherins) and protamine-histone exchange factors.

Recently, genetic determinants of CI from *Wolbachia*, called CI factors or Cifs, have been identified (*Beckmann et al., 2019a*; *Beckmann and Fallon, 2013*; *Beckmann et al., 2017*; *Chen et al., 2019*; *Shropshire et al., 2018*; *LePage et al., 2017*; *Cooper et al., 2019*; *Meany, 2018*; *Lindsey et al., 2018*; *Gillespie et al., 2018*). The Cif proteins are encoded by two-gene operons (*Beckmann and Fallon, 2013*; *Lindsey et al., 2018*) that are commonly found within *Wolbachia* prophage (WO phage) regions termed eukaryotic association modules (EAMs) (*LePage et al., 2017*; *Bordenstein and Bordenstein, 2016*). The *cif* genes, however, have been traced to more ancient bacterial plasmids (*Gillespie et al., 2018*). Moreover, the EAM found in WO phages derives from a *Rickettsial* plasmid (*Gillespie et al., 2012*). The *cif* genes themselves are found in *Wolbachia*, *Rickettsia*, and *Orientia* (*Gillespie et al., 2018*), but *Orientia* and *Rickettsia* generally lack phage. In sum, the *cif* family is diverse and predates *Wolbachia* and its phages (*Beckmann et al., 2019a*; *Gillespie et al., 2018*).

The downstream genes in the *cif* operons, *cidB* or *cinB*, encode enzymatic activities essential to their ability to induce CI when expressed in the germlines of transgenic flies (*Beckmann et al., 2019a*). They have either deubiquitlyase (*Beckmann et al., 2017*) (Ci<u>d</u>) or nuclease (*Chen et al., 2019*) (Ci<u>n</u>) enzymatic functions. The corresponding upstream genes, *cidA* or *cinA*, encode proteins that bind tightly to CidB and CinB proteins, respectively, from the same operon. Dual transgenic expression in *Drosophila melanogaster* of *Wolbachia* CidA and CidB proteins precisely mimics natural CI (*Beckmann and Fallon, 2013*; *Beckmann et al., 2017*; *LePage et al., 2017*). We have modeled CI as a toxin-antidote (TA) system with CidB as the toxin and CidA the antidote (*Beckmann et al., 2019a*; *Poinsot et al., 2003*; *Hurst, 1991*; *Bourtzis et al., 2003*; *Beckmann et al., 2019b*). We annotate Cifs with superscripts identifying the *Wolbachia* strain of origin; for instance, CidB$^{wPip}$ is the toxin from the *w*Pip *Wolbachia* endosymbiont of *Culex pipiens* (*Beckmann et al., 2019a*).

In toxin-antidote (type II) systems in free-living bacteria, toxin and antidote proteins are translated together and bind directly to one another (*Yamaguchi et al., 2011*). Toxicity occurs if cells no longer synthesize the proteins because the antidote protein is degraded much more rapidly than the toxin, thereby releasing active toxin. We posit that CifA and CifB proteins behave similarly (*Beckmann et al., 2019a*). Not only do CidA and CidB bind together in a cognate-specific manner, but CidA$^{wPip}$ coexpression also suppresses CidB$^{wPip}$ toxicity in yeast (*Beckmann et al., 2017*). The antidote role of CidA has been inferred from bi-directional crosses among infected *C. pipiens*

mosquitoes (*Bonneau et al., 2018*) and from transgenic *Drosophila* CidA$^{wMel}$ and CinA$^{wPip}$ experiments (*Chen et al., 2019*; *Shropshire et al., 2018*). As in natural CI, the incompatibility induced by transgenic *cidAB$^{wMel}$* can be rescued by a cognate maternal *Wolbachia* infection (*LePage et al., 2017*). Conversely, natural *w*Mel-induced CI can be rescued by transgenic overexpression of CidA$^{w}$-$^{Mel}$ in mothers (*Shropshire et al., 2018*). In transgenic Cid models, incompatibility depends on the CidB deubiquitylase (DUB) activity (*Beckmann et al., 2017*). Post-translational ubiquitin modifications alter protein stability, localization, and interactions (*Hochstrasser, 1996*; *Ronau et al., 2016*). Active site (C1025A) mutation of CidB$^{wPip}$ eliminates CI and CI-like cytology in transgenic insects (*Beckmann et al., 2017*). CidB DUB targets are unknown.

Here we focus on identification of CidB targets using both physical and genetic interaction screens. We use yeast and transgenic *Drosophila* to identify dosage suppressors of CidB-derived toxicity. Identification of suppressors of CI may be important beyond aiding in elucidation of CI mechanisms. CI suppression could weaken world-wide *Wolbachia*-based mosquito control efforts and reduce *Wolbachia* equilibrium frequency. Host genes can modulate *Wolbachia*'s reproductive phenotypes (*Hornett et al., 2006*; *Metcalf et al., 2014*; *Bordenstein et al., 2003*; *Cooper et al., 2017*; *Reynolds and Hoffmann, 2002*), and natural selection favors host suppression of CI (*Turelli, 1994*). CI is weak in some host insects, (*Cooper et al., 2017*; *Conner et al., 2017*; *Hamm et al., 2014*) but in others is strong (*Poinsot et al., 2003*; *Merçot and Charlat, 2004*). A CI-inducing *Wolbachia* strain can change its kill ratio in heterologous hosts (*Walker et al., 2011*; *Bordenstein et al., 2003*).

We identify karyopherin-α (Kap-α/importin-α), as both a dosage suppressor of CidB toxicity and a CidB binder. Kap-α is a conserved nuclear-import receptor for proteins with classical nuclear localization signals (NLSs) (*Chen and Madura, 2014*). After substrate recruitment, Kap-α associates with karyopherin-β and escorts cargo through nuclear pores (*Chook and Blobel, 2001*). Nuclear Ran-GTP binding releases the cargo, and the karyopherins recycle to the cytoplasm (*Goldfarb et al., 2004*). CidB-Kap-α interaction connects CI induction and nuclear transport. Our study also highlights CidB association with protamine-histone exchange chaperones P32 and Nap1. Importantly, cognate CidA antidote binding to the CidB toxin eliminates these interactions. These discoveries identify the first potential CI molecular targets that comport with prior cytological observations (*Ferree and Sullivan, 2006*; *Landmann et al., 2009*).

## Results

### Host background modulates CidB toxicity in yeast

Ectopic expression of CidB$^{wPip}$ in *Saccharomyces cerevisiae* causes strong temperature-sensitive growth inhibition (*Beckmann et al., 2017*). We sought to identify yeast factors modulating CidB$^{wPip}$ toxicity. As a first step, we determined whether yeast host background altered toxicity of CidB$^{wPip}$ and also tested other *Wolbachia* CI toxins for growth inhibition. Different CI-inducing *Wolbachia* strains have distinct Cif repertoires (we follow *Beckmann et al., 2019a* in using the Cif term to designate any general CI factor). We previously distinguished three biochemical toxin types (*Beckmann et al., 2019a*). C<u>id</u> toxins are <u>DUB</u>s, C<u>in</u> toxins are predicted <u>nuclease</u>s, and C<u>nd</u> toxins have both <u>nuclease</u> and <u>DUB</u> domains (*Beckmann et al., 2019a*; *Beckmann et al., 2017*; *Gillespie et al., 2018*).

When expressing *cidB$^{wPip}$* in two different yeast backgrounds, BY4741 (*Brachmann et al., 1998*) and W303-1A (*Thomas and Rothstein, 1989*), we noticed greater sensitivity to its expression in W303-1A (*Figure 1a*). Differential sensitivity was also observed with expression of two previously uncharacterized toxin alleles, *cidB$^{wHa}$* (*w*Ha infects *D. simulans*) and *cndB$^{wStr}$* (*w*Str infects planthoppers). $^{FLAG}$*cidB$^{wPip}$* had the strongest toxicity, and $^{FLAG}$*cndB$^{wStr}$* (truncated after the DUB domain) the weakest. Protein levels of $^{FLAG}$CndB$^{wStr}$ showed that variance in toxicity was not simply attributable to differences in protein expression (*Figure 1—figure supplement 1*). Our previous study demonstrated cognate-specific rescue with two *cif* operons (*Beckmann et al., 2017*). Here we observed that the Cid-class operon from *w*Ha also showed cognate-specific rescue (*Figure 1b*). When *cidB$^{wHa}$* was co-expressed with non-cognate *cifA* genes, either no rescue or even enhanced toxicity was seen. Expression of the CifA proteins alone induced no growth defects (*Figure 1c*, *Figure 1—figure*

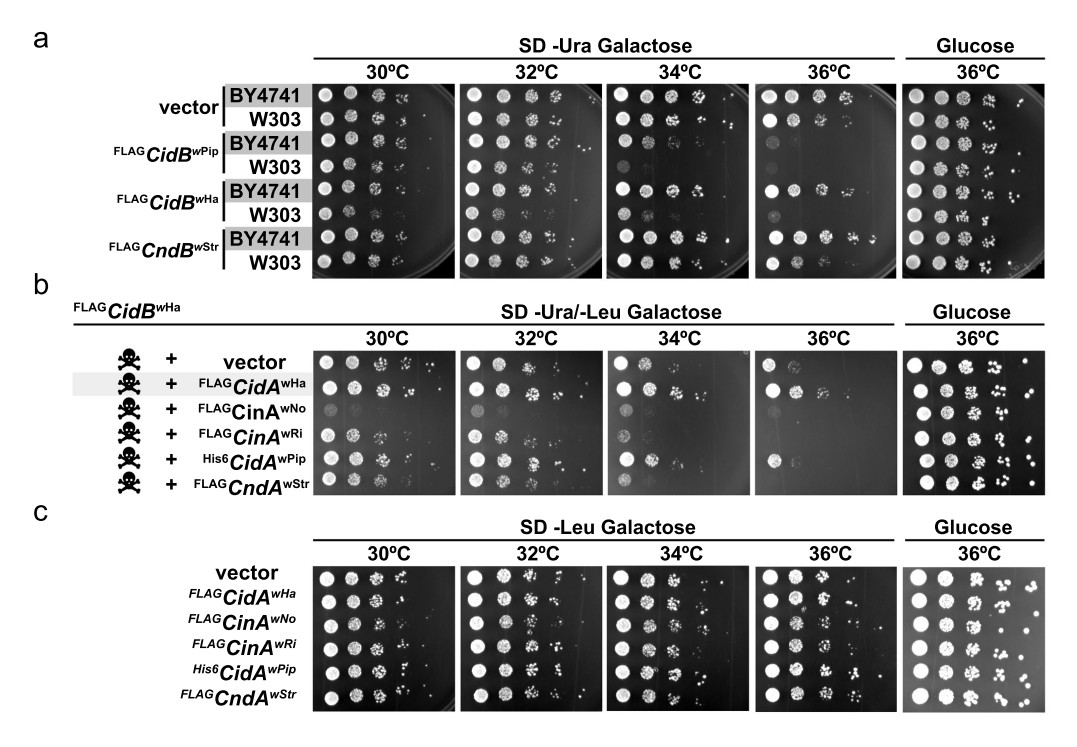

**Figure 1.** Cif toxicity in *S. cerevisiae*. (**a**) Five-fold dilutions of yeasts BY4741 and W303-1A carrying galactose-inducible epitope-tagged *Wolbachia* genes on pRS416GAL1. Three Cif homologs from *Wolbachia* strains *w*Pip, *w*Ha, and *w*Str showed strong to mild toxicity. All three showed increased toxicity in W303-1A compared to BY4741 (three replicates). (**b**) Toxin-antidote behavior was exhibited by the *cidAB*$^{wHa}$ operon. $^{FLAG}$CidB$^{wHa}$ exhibited toxicity at 36˚C when expressed from pRS416GAL1. Co-expression of cognate partner $^{FLAG}$CidA$^{wHa}$ from the 2-micron plasmid pRS425GAL1 rescues growth. Non-cognate partners did not rescue. Conversely, expression of $^{FLAG}$CinA$^{wNo}$ from a bidirectionally incompatible *Wolbachia* strain *w*No, enhanced toxicity of $^{FLAG}$CidB$^{wHa}$ (four replicates). (**c**) CifA expression alone was nontoxic (three replicates).

The online version of this article includes the following figure supplement(s) for figure 1:

**Figure supplement 1.** Expression Analysis of CI Factors in Yeast.

supplement 1b). These data suggest that yeast genetic background modulates Cif toxicity. This is congruent with observed variance in toxicity in natural CI within insects.

## Yeast dosage suppressors of cidB toxicity

The poor relative growth of the W303-1A yeast strain in the presence of CidB might reflect strain-specific differences in the activities of targets or mediators of CidB toxicity. For example, there may be lower levels in W303-1A compared to BY4741 of a key ubiquitin-protein conjugate that is essential for growth and targeted by the CidB DUB. We therefore sought to identify yeast genes from a high-copy, tiled genomic library that were capable of suppressing *cidB*$^{wPip}$ toxicity (*Jones et al., 2008*). After rescreening the initial set of isolates (*Supplementary file 1a*; *Figure 2—figure supplements 1* and *2*), seven library plasmids showed suppression of toxicity (*Figure 2a–b*). A plasmid with the endogenous *URA3* gene served effectively as a positive control for plasmid coverage in the screen inasmuch as yeast transformants with this plasmid no longer needed to retain the *URA3*-based $^{His6}$*cidB*$^{wPip}$ plasmid to grow on plates lacking uracil. The *URA3* plasmid was identified 16 times, suggesting ~16 fold library coverage.

Individual genes from each genomic insert were subcloned to identify the responsible suppressor. Plasmids with *SRP1*, *RTT103*, *HRP1*, or *FET4* alone suppressed *cidB*$^{3xFLAG-wPip}$-induced toxicity (*Figure 2b,c*); *FET4* was the only gene on the weakly suppressing YGPM32b05 plasmid, so it was not subcloned further (*Figure 2a*). To rule out suppression of CidB$^{3xFLAG-wPip}$ protein levels by the high-copy plasmids, we overexpressed *SRP1* and *RTT103* in yeast cotransformants and found no reduction of CidB$^{3xFLAG-wPip}$ (*Figure 2d*). Based on the incomplete *RTT103* sequence present on one

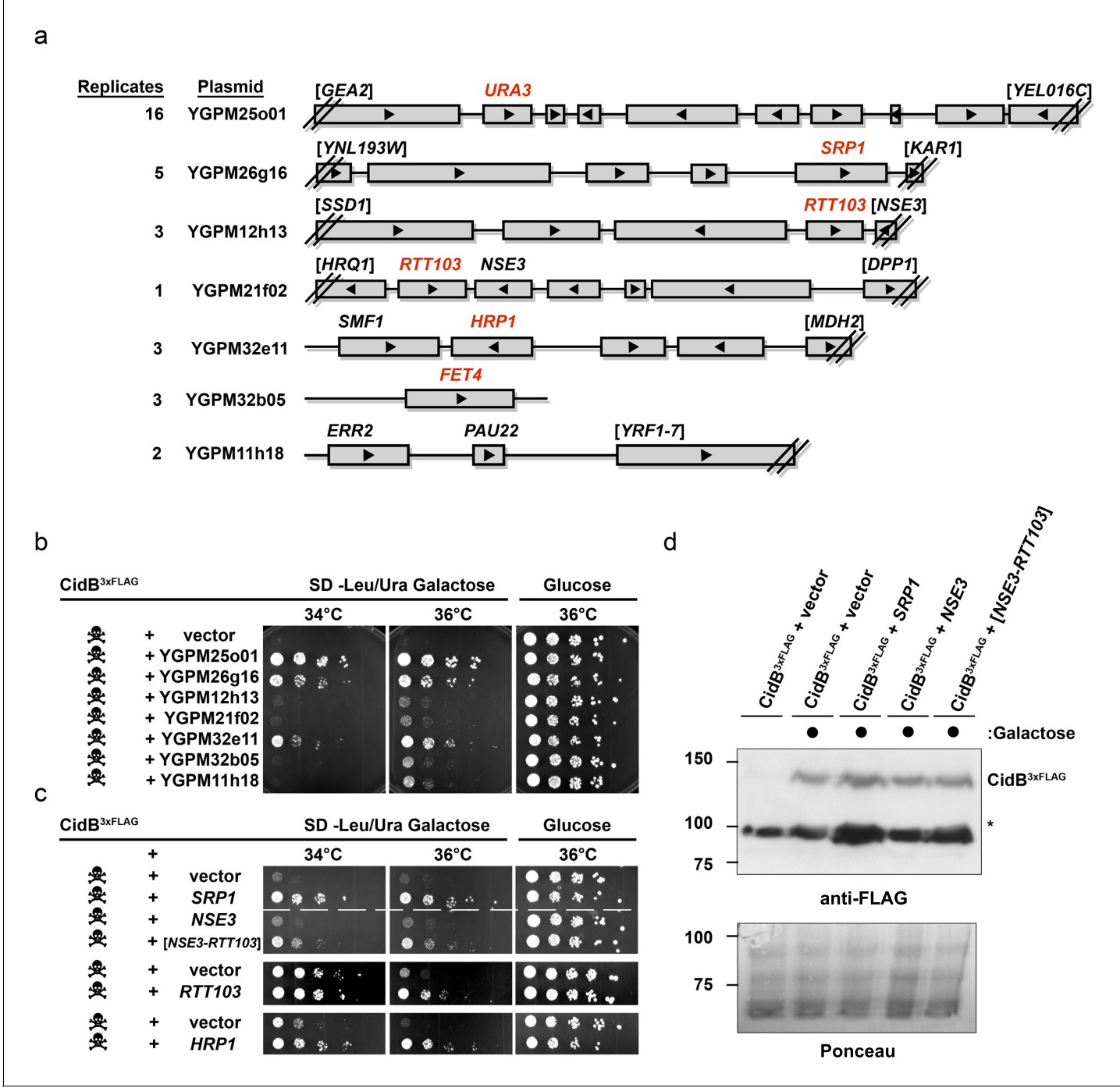

**Figure 2.** Yeast Suppressors of CidB. (**a**) Seven library plasmids were high-copy suppressors of CidB[wPip] toxicity. Red genes suppressed when individually sub-cloned. Library plasmid YGPM25o01 includes *URA3* and measures screen efficiency since it is an expected suppressor; Backslashes and brackets denote ORF truncations. (**b**) Five-fold serial dilutions of yeast (W303-1A) with recovered suppressing library plasmids co-transformed with pRS416GAL1-CidB[3xFLAG-wPip]. Library plasmid suppression varied. Suppression by YGPM25o01 (*URA3* control), YGPM26g16, and YGPM32e11 was strong and consistent (three replicates). Plasmids YGPM12h13, YGPM21f02, YGPM32b05, and YGPM11h18, showed weaker and less consistent suppression across four replicates. (**c**) Individual yeast genes *SRP1*, *RTT103*, and *HRP1* suppressed CidB[wPip] toxicity (three replicates). (**d**) Immunoblot analysis confirmed that suppressor plasmids do not reduce CidB expression. CidB and suppressors were controlled by *GAL1* and endogenous promoters, respectively. Asterisk, an unknown cross-reacting yeast protein. Ponceau S staining indicated relative sample loading.

The online version of this article includes the following figure supplement(s) for figure 2:

**Figure supplement 1.** Eliminating false positives from the high-copy [His6]CidB[wPip] suppression screen.

*Figure 2 continued on next page*

Figure 2 continued

**Figure supplement 2.** Five-fold serial dilutions of yeast (BY4741) with recovered suppressing library plasmids co-transformed with pRS416GAL1-$^{His6}$CidB.

suppressing plasmid, codons 335–409 were sufficient for suppression (*Figure 2c*). *HRP1* encodes an RNA-binding protein involved in processing the 3'-ends of mRNA precursors and mRNA export (*Kessler et al., 1997*). *RTT103* is a transcription termination factor for RNA polymerase II (*Nemec et al., 2017*). *FET4* is an iron transporter (*Dix et al., 1997*). *SRP1/KAP60* encodes the yeast karyopherin-α protein (*Loeb et al., 1995*). *SRP1* was the most robust suppressor (followed by *HRP1*); thus, we focused on this gene.

## SRP1 suppression of CidB relies on nuclear import

Srp1 has functions beyond nuclear import (*Chen and Madura, 2014*). Specific functions can be differentially inactivated by specific point mutations. The mutation S116F (*srp1-31* allele) disrupts binding between Srp1 and substrate NLS elements, while the E145K mutation (*srp1-49* allele) inhibits its function in co-translational protein degradation (*Chen and Madura, 2014*; *Loeb et al., 1995*). Only the NLS-binding mutation (S116F) impaired the ability of high-copy *SRP1* to suppress *cidB*$^{wPip}$

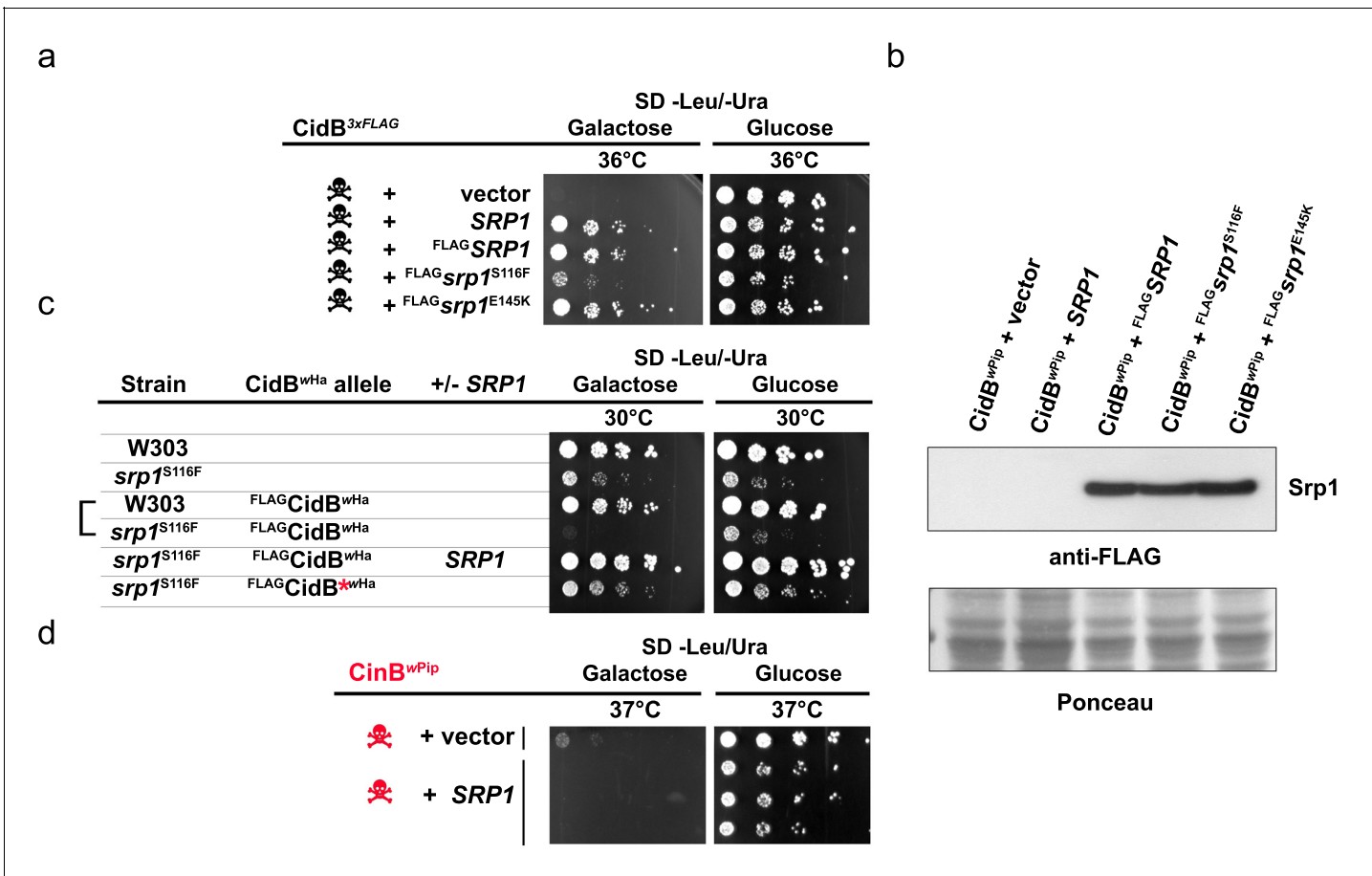

**Figure 3.** Analysis of high-copy *SRP1* suppression of CidB$^{wPip}$ toxicity in yeast. (a) Differential impact of mutations affecting distinct Srp1 functions. An *srp1* mutation impairing NLS binding (S116F) weakened suppression in W303-1A. E145K, which inhibits cotranslational protein degradation, did not impact suppression (three replicates). (b) Immunoblot analysis showed equivalent protein levels in *srp1* mutants. Ponceau S staining demonstrated similar loading (three replicates). (c) The *srp1-S116F* mutation sensitized W303-1A yeast to $^{FLAG}$CidB$^{wHa}$-induced toxicity in 6/7 replicates. Wild-type *SRP1* complemented the mutation (5th row). Red * indicates an inactive DUB catalytic mutant control (6th row). Blank columns are empty vectors. (d) High-copy *SRP1* did not suppress CinB$^{wPip}$ toxicity in BY4741 yeast (three replicates).

toxicity (*Figure 3a*). All Srp1 proteins were expressed similarly, so variance in protein abundance cannot account for variation in suppression (*Figure 3b*). Conversely, the *srp1-S116F* mutation in the lone chromosomal copy of *SRP1* increased *cidB* toxicity; this synthetic growth defect was most clearly seen with the *cidB*^wHa^ toxin (*Figure 3c*). High-copy *SRP1* suppression appeared to be specific to Cid (DUB) toxins insofar as *SRP1* did not suppress the growth impairment caused by the *cinB*^wPip^ paralog (*Figure 3d*). These results demonstrate a functional link between *cidB* toxicity and nuclear protein import in yeast.

## CidB-*Drosophila* protein interactome implicates nuclear transport and nucleosome assembly

We created a recombinant expression construct for purification of a catalytically inactive *w*Pip ^His6^-CidB derivative bearing a C1025A active-site mutation (*Beckmann et al., 2017*). Similarly inactivated DUBs often bind substrates more tightly than their wild-type counterparts (*Morrow et al., 2018*). This protein, ^His6^CidB*, was expressed in *E. coli* and bound to a cobalt-affinity resin. Lysates from adult *D. melanogaster* flies (both sexes) were passed over the ^His6^CidB* resin. Enriched proteins were eluted and identified by liquid chromatography-tandem mass spectrometry (LC-MS/MS). Samples were compared to eluates from mock control columns lacking ^His6^CidB* (*Figure 4a*). From two biological replicates, 169 proteins were enriched on ^His6^CidB* (*Supplementary file 1b*); this was reduced to 45 proteins based on biological triplicates (*Figure 4b*; *Supplementary file 1c*). We classified these top hits into functional categories (*Figure 4—figure supplement 1*) (*Baldridge et al., 2017*). The largest functional category from the screen was 'ribosome structure/biogenesis/translation' with 31% of hits. The second largest category was 'DNA replication/repair/packaging/cell division' with 13%. To identify the most robust hits, we subjected these raw data to peptide spectral analysis (described in Materials and methods).

The refined CidB* interactome is given in *Table 1a*. Eluates were enriched for ubiquitin, which served as a positive control for the DUB substrate trap. The top two hits with the strongest peptide frequency values were both karyopherins, Kap-α2 (a karyopherin-α ortholog of yeast *SRP1*) and Moleskin/Imp-7, a karyopherin-β paralog (which does not associate with karyopherin-α). The ^His6^-CidB* resin also enriched two proteins that function in protamine removal and nucleosome assembly, P32/TAP and, to a lesser degree, Nap1.

## CidA scrambles the CidB-*Drosophila* protein interactome

CidA factors bind specifically to cognate CidB proteins and suppress CidB or *Wolbachia* toxicity in the yeast and insect CI models (*Beckmann et al., 2017*; *Shropshire et al., 2018*). We hypothesized that CidA 'rescue' is due to CidA association changing CidB interactions with its substrates or cofactors. To test this idea, we repeated the substrate-trap experiments with ^His6^CidB* bound to the cognate ^FLAG^CidA from *w*Pip. With two biological replicates, 239 proteins were enriched on the CidA-CidB* column (*Supplementary file 1d*); this was reduced to 67 proteins following analysis of a third replicate (*Supplementary file 1e*; *Figure 4b*). These top hits were also sorted into functional categories (*Supplementary file 1e*; *Figure 4—figure supplement 2*). In support of the hypothesis that CidA disrupts the CidB interactome, the proteins bound to CidB* alone and to the CidA-CidB* complex were completely different except for three proteins, Dek, Non2, and BSF (*Figure 4b*; *Supplementary file 1c* cross referenced with *Supplementary file 1e*). Importantly, the nuclear transporters Kap-α2 and Moleskin as well as the histone chaperones P32 and Nap1 were no longer enriched when CidA^wPip^ was bound to CidB^wPip^. These data are consistent with CidA blocking CidB access to its substrates or cofactors.

## The CidA-*Drosophila* protein interactome

We also identified a *Drosophila* protein interactome for ^His6^CidA^wPip^ by itself (*Figure 4b*; *Supplementary file 1f and 1g*; *Figure 4—figure supplement 3*). CI-relevant targets of the CidB DUB would be predicted to be absent. Indeed, none of ^His6^CidA-*Drosophila* protein interactions identified were part of the CidB* or CidA-CidB* interactomes (cross referencing *Supplementary file 1g, 1e and 1c*). Surprisingly few robust ^His6^CidA interactions were identified (*Table 1c*). The three statistically significant hits were a predicted nucleotide exchange factor Roe1, a lipid kinase Pi3K92E, (*Leevers et al., 1996*) and aminolevulinic acid synthase, Alas (*de Mena et al., 1999*).

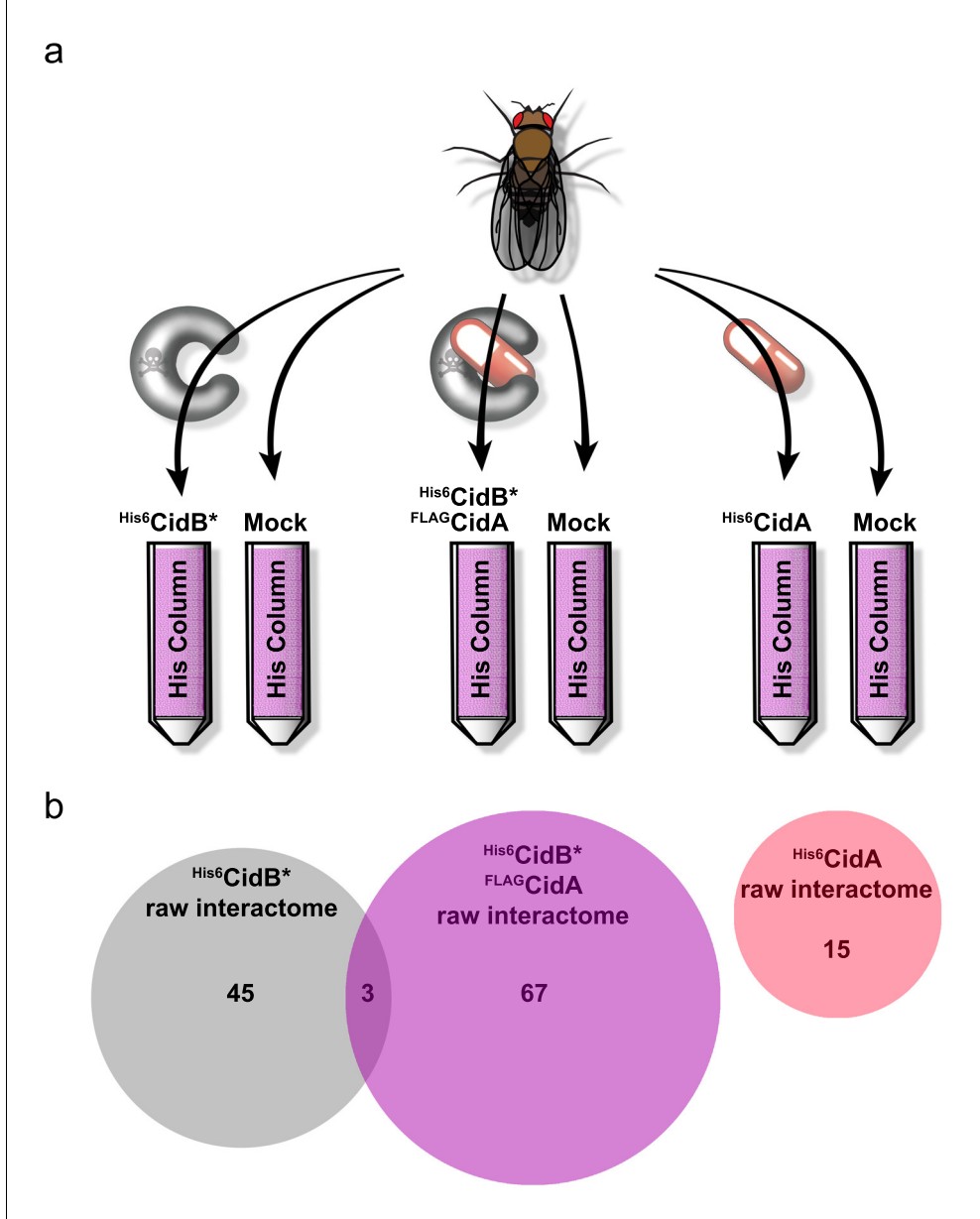

**Figure 4.** Drosophila Interactome Analysis. (**a**) Experimental pipeline for defining CidA and CidB interactomes. Soluble lysates from *Drosophila* adults were passed over columns bound to the indicated recombinant proteins and washed. Remaining proteins were eluted and subjected to in-solution LC-MS/MS analysis. (**b**) Venn diagram of protein identifications from raw biological triplicate measurements. The $^{His6}$CidB* interactome was dramatically changed when it was bound to $^{FLAG}$CidA. The interactome of $^{His6}$CidA itself was modest and showed no overlap with the *Drosophila* proteins bound to either CidB* or the CidA-CidB* complex.

The online version of this article includes the following figure supplement(s) for figure 4:

**Figure supplement 1.** Triplicate Enrichment Interactome for $^{His6}$CidB*$^{wPip}$.
**Figure supplement 2.** Triplicate Enrichment Interactome for the $^{FLAG}$CidA$^{wPip}$/$^{His6}$CidB*$^{wPip}$ complex.
**Figure supplement 3.** Triplicate Enrichment Interactome for $^{His6}$CidA$^{wPip}$.

Whether any of these interactions is relevant to CI physiology is unknown. It is possible that CidA has few strong interactions with host proteins by itself, with its main function being tight association with CidB in order to remodel the latter's protein interactome.

**Table 1.** Final refined interactomes of CidB*[wPip], CidB*/CidA[wPip], and CidA[wPip] ranked by F-Score.

Peptide spectral matches (PSM) of the top enriched proteins are reported. PSMs are reported as the average of three biological replicates, each a summation of 2 technical replicates; (six total samples, three biological replicates). Mock is an *E. coli* negative control without plasmid. P-values were calculated by two sample T-test assuming unequal variances of the replicates. Ubiquitin served as an intrinsic positive control.

**a** [His6]CidB* Interactome

| Protein | kDa | UniProt | F-Score | CidB* PSM | Mock PSM | p-value |
|---|---|---|---|---|---|---|
| Kap-α2 | 58 | IMA_DROME | 1.00 | 10.7 | 0 | 0.004 |
| Moleskin (Kap-β) | 119 | Q9VSD6_DROME | 0.85 | 36 | 7.7 | 0.042 |
| Modulo | 60 | A0A0B4K7G4_DROME | 0.83 | 53.3 | 11.3 | 0.048 |
| P32 | 29 | Q7JXC4_DROME | 0.76 | 21 | 2.7 | 0.036 |
| Vitellogenin-2 | 50 | VIT2_DROME | 0.54 | 37.3 | 20 | 0.092 |
| Cdep | 132 | A0A0C4DHA1_DROME | 0.48 | 14.3 | 9 | 0.121 |
| l(3)72Ab | 245 | U520_DROME | 0.47 | 22.7 | 5.7 | 0.051 |
| 14-3-3zeta | 28 | A0A0B4KEH0_DROME | 0.45 | 5 | 2 | 0.015 |
| Ubiquitin | 18 | RS27A_DROME | 0.44 | 6.7 | 3 | 0.065 |
| Nap1 | 43 | Q9W1G7_DROME | 0.21 | 20.3 | 11 | 0.122 |

**b** [His6]CidB* + [FLAG]CidA Interactome

| Protein | kDa | UniProt | F-Score | CidB*/A PSM | Mock PSM | p-value |
|---|---|---|---|---|---|---|
| Pkcdelta | 207 | Q9VYN1_DROME | 0.91 | 14.3 | 2 | 0.01 |
| TfIIFalpha | 64 | T2FA_DROME | 0.82 | 12.7 | 2.7 | 0.024 |
| La-related | 161 | Y1505_DROME | 0.80 | 7.3 | 2.3 | 0.081 |
| Bunched | 125 | BUN2_DROME | 0.79 | 8 | 1.7 | 0.013 |
| AP-3 subunit beta | 127 | Q9W4K1_DROME | 0.74 | 60 | 19 | 0.012 |
| AP-3 subunit delta | 115 | AP3D_DROME | 0.71 | 56.7 | 16.3 | 0.002 |
| Sals | 101 | Q58CJ5_DROME | 0.67 | 28 | 14 | 0.127 |
| CG4069 | 56 | Q9VTZ7_DROME | 0.65 | 25.3 | 5.7 | 0.003 |
| Ssrp | 82 | SSRP1_DROME | 0.65 | 43 | 18 | 0.014 |
| Chrac-14 | 14 | Q9V444_DROME | 0.65 | 4 | 2 | 0.058 |
| Dre4 | 128 | SPT16_DROME | 0.63 | 62.3 | 22 | 0.001 |
| AP-3mu | 47 | O76928_DROME | 0.61 | 18.7 | 4.7 | 0.01 |
| Shaggy | 78 | A8JUV9_DROME | 0.58 | 27 | 11.3 | 0.025 |
| CG2025-RA | 133 | Q9VYT3_DROME | 0.55 | 80 | 43.7 | 0.07 |
| Mical | 526 | A0A0B4K703_DROME | 0.48 | 39.3 | 20.7 | 0.01 |
| Bsf | 157 | Q9VJ86_DROME | 0.45 | 187 | 105 | 0.03 |
| Purple | 19 | PTPS_DROME | 0.32 | 33 | 21.7 | 0.007 |
| CG11444 | 23 | Q9W4J4_DROME | 0.26 | 12.3 | 8.3 | 0.014 |

**c** [His6]CidA Interactome

| Protein | kDa | UniProt | F-Score | CidA PSM | Mock PSM | p-value |
|---|---|---|---|---|---|---|
| Roe1 | 24 | GRPE_DROME | 0.97 | 6 | 0 | 0 |
| Pi3K92E | 127 | P91634_DROME | 0.36 | 4 | 2.3 | 0.021 |
| CG17271 | 33 | Q9VDI5_DROME | 0.30 | 10 | 7 | 0.095 |
| Alas | 59 | O18680_DROME | 0.29 | 10.7 | 6.3 | 0.043 |
| CG6984 | 31 | Q7K1C3_DROME | 0.10 | 8.3 | 6.3 | 0.07 |

## Overexpressed *Drosophila* karyopherins and protamine-histone chaperones suppress CI

Because both the yeast CidB suppressor screen and the *Drosophila* interactome screen identified karyopherin-α, we determined whether increased dosage of karyopherin-α genes might suppress CI in fruit flies, similar to observed results in yeast. *D. melanogaster* has four paralogous karyopherin-α genes (α1, α2, α3, α4) (*Phadnis et al., 2012*; *Pieper et al., 2018*). Two of them, Kap-α1 and Kap-α 2, were chosen because the first is the closest in sequence to yeast *SRP1* and the second was the top CidB* interactome hit. In order to test these genes for CI suppression, we switched operons from *cidAB*$^{wPip}$ to *cidAB*$^{wMel}$ as *cidAB*$^{wPip}$ is too toxic and kills all embryos resulting from transgenic male flies. The *cidAB*$^{wMel}$ operon effect is weaker. Suppression in fruit flies was expected to be incomplete, and the CI effects must be weak enough to detect suppression. *w*Mel is also native to *D. melanogaster*.

We first optimized transgenic *cidAB*$^{wMel}$-induced CI under the Gal4/UAS system. CI induction was strongly temperature dependent (*Figure 5a*; *Supplementary file 1h*). Although higher growth temperatures caused greater reductions in egg hatch rates, we found 22°C was the optimal temperature for observing partial suppressive effects on CI (*Figure 5b*). Expression of either *Drosophila* karyopherin-α paralog in the female germline partially suppressed CI caused by transgenic expression of CidA-B$^{wMel}$ in males; yeast *SRP1* did not. When transgenic GFP was used as a negative control, however, it also caused a partial suppression that was not statistically distinguishable from the karyopherin-α suppression. We tried to boost maternal expression in order to increase the relative magnitude of the karyopherin-α effects, but when we switched to the stronger maternal triple driver (MTD), karyopherin-α overexpression in females caused embryos to die independent of CI (*Figure 5c*). Thus, the data in *Figure 5b* must be interpreted in the context of *Figure 5c*. The suppression effects were significant but relatively small because we utilized a weaker driver (NGT) to limit maternal toxicity.

We next tested whether maternal overproduction of P32 or Nap1 (also identified as a potential CidB substrates or cofactors; *Table 1*) could suppress transgenic CI and found that overexpression of P32 showed highly significant suppression relative to the GFP control, increasing egg hatch rates by ~30% (*Figure 5b*). Suppression was equivalent to the rescue observed with transgenic expression of the actual CidA antidote. Importantly, when we measured the suppressive effects of karyopherin-α and P32 overexpression in the female germline in matings with male flies carrying *w*Mel bacterial infections, partial but highly significant suppression was observed for both P32 and karyopherin-α but not for GFP (*Figure 5d*). We conclude that the suppression by karyopherin-α and P32 was relevant to natural CI and that the weak suppression by GFP was an artifact of the transgenic CI induction model.

## Discussion

The *Wolbachia* CidA and CidB proteins were recently found to be central to CI, but no CidB targets were known. Two orthogonal screens of CidB genetic and physical interactions in *S. cerevisiae* and *D. melanogaster*, respectively, identified the nuclear-import receptor karyopherin-α (Kap-α). Kap-α bound to CidB and genetically suppressed CidB-derived defects when overexpressed. The Kap-α NLS-binding site was required for suppression of CidB toxicity. CidB also binds *Drosophila* P32 and Nap1, which promote protamine-histone exchange. Overexpression of either Kap-α or P32 in female insect germlines suppressed natural CI. We also show that CidA in mother flies is sufficient to rescue both transgenic and wild CI.

Notably, CidB-associated proteins such as Kap-α, P32, and Nap1 disappear when affinity purifications are performed in the presence of the antidote CidA. Instead, the CidA-CidB heterodimer has robust interactions with a number of other proteins, many that are not in the nucleus. For example, three of the four subunits of the AP-3 clathrin adaptor complex were identified; AP-3 regulates vesicle trafficking to lysosomes (*Park and Guo, 2014*). These interactions of the complex might tether CidB at sites away from nuclear CI induction targets. The dramatic changes in the CidB-*Drosophila* protein interactome if CidB is bound to CidA suggest the rescue function of CidA acts through alteration of CidB localization or access to key target proteins.

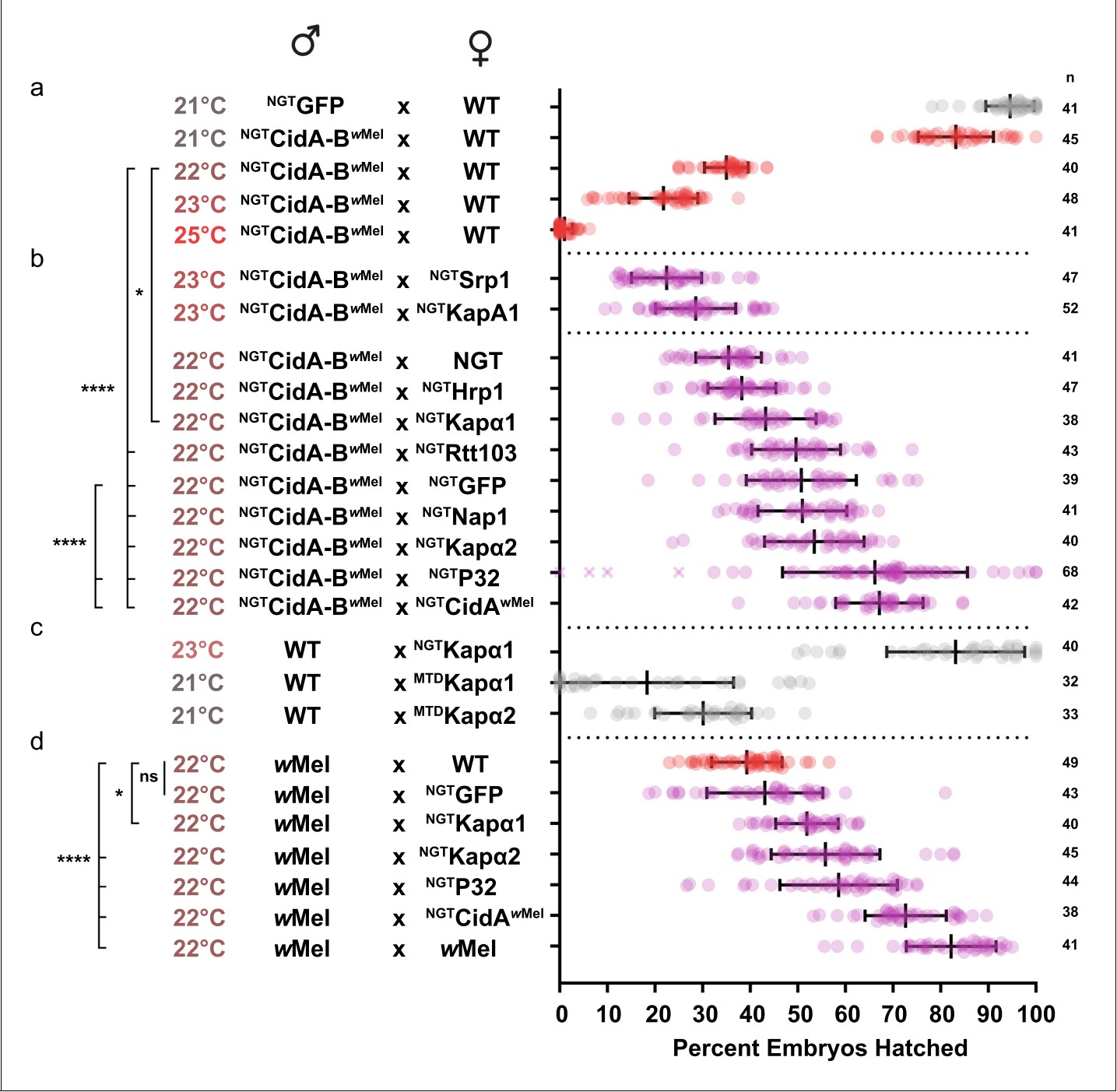

**Figure 5.** Suppression of CI in *Drosophila*. (**a**) Transgenic CI was temperature sensitive. (**b**) Yeast *SRP1* and *HRP1* did not suppress CI in *Drosophila* and serve as negative controls. At 22˚C, overexpression of *D.m.*Kap-α1, *S.c.*Rtt103, GFP, *D.m.*Nap1, *D.m.*Kap-α2, *D.m.*P32 and CidA*wMel* suppressed transgenic CI relative to the control. Both *D.m.*P32 and CidA*wMel* suppression were still highly significant when compared to the GFP control. (**c**) CI suppressive effects of karyopherin overexpression were countered by its maternal toxicity. (**d**) *D.m.* Karyopherins and *D.m.*P32 significantly suppressed bacterial (*w*Mel) CI; GFP did not. Error bars represent means ± s.d. *p<0.05, **p<0.01, ****p<0.0001 by ANOVA with multiple comparison between all groups and Tukey's post-hoc analysis; four outliers (x) removed by ROUT analysis.

The online version of this article includes the following figure supplement(s) for figure 5:

**Figure supplement 1.** PCR analysis demonstrates that transgenic flies used in this study are not infected with *Wolbachia*.

## Comparison of model fly and yeast CI systems

An important question regarding Cid and Cin growth effects in yeast is whether the observed toxicity and suppression occur through mechanisms similar or identical to CI induction and rescue in insects. To date, our data show a striking concordance between the yeast and *Drosophila* analyses, suggesting mechanistic insights into CI can indeed be inferred from yeast studies. First, different yeast strain backgrounds diverge markedly in their sensitivity to the CidB toxin, similar to the wide differences in CI penetrance among various insect host strains (*Cooper et al., 2017*; *Reynolds and Hoffmann, 2002*; *Merçot and Charlat, 2004*). Second, the specific suppression by the CidA$^{wHa}$ antidote of the CidB$^{wHa}$ toxin in yeast lends further support to the cognate specificity predicted from previous analysis of *cif* operons (*Beckmann et al., 2017*; *Beckmann et al., 2019a*; *Beckmann et al., 2019b*). The enhanced toxicity caused by noncognate CidA factors when coexpressed with CidB$^{wHa}$ may also contribute to incompatibilities in natural populations, although we do not know the mechanism. Third, CidA suppression of CidB toxicity ('rescue') in yeast is now paralleled by analogous observations in transgenic flies (*Figure 5*). Females expressing CidA$^{wMel}$ alone suppress the incompatibility of males with either transgenic *cidAB$^{wMel}$* operons or *w*Mel infections. This supports and extends previous studies that demonstrated similar rescue effects with transgenic expression of CidA$^{wMel}$ or CinA$^{wPip}$ in females (*Chen et al., 2019*; *Shropshire et al., 2018*; *Shropshire and Bordenstein, 2019*). Finally, overexpression of Kap-α both suppressed CidB toxicity in yeast and suppressed *w*Mel-induced CI in flies. We conclude that CI targets conserved pathways in the *S. cerevisiae* and *Drosophila* models.

The fact that GFP weakly suppressed transgenic CI but not wild CI suggests that experiments utilizing the Gal4/UAS systems can produce nonspecific suppression. Cytological studies suggest that CI might have multiple stages. Most embryos die following the first zygotic nuclear division, but escapees die at later stages (*Beckmann et al., 2017*; *Callaini et al., 1996*). One possible explanation of the weak nonspecific suppression could be that crosses to mothers with a UAS-driven transgene alleviate a later, secondary stage of CI killing by reducing embryonic expression of the transgenic CidB toxin in older embryos. This might result from binding of the Gal4 transcription factor to the maternal UAS insertions, titrating it from the transgenic *UAS-cidB* gene.

## Mechanistic models of CI induction and rescue

It is not yet clear if the top hits in our screens, such as Kap-α and P32, are deubiquitylated by the CidB enzyme or how this could help account for their functions in CI. Srp1/Kap-α (and Hrp1, another top hit) are known to be ubiquitylated in yeast based on proteomic surveys (*Swaney et al., 2013*). One highly speculative model invokes CidB cleavage of ubiquitin from both Kap-α and histone chaperones such as P32 (or the histones themselves), reducing their functionality. Histone H2A and H2B are well characterized as ubiquitylated proteins, and histone H2B was identified in our CidB\*-binding screen (*Figure 4—figure supplement 1*). Its ubiquitylation may promote histone H3.3 loading and nucleosome formation. There is evidence for ubiquitin-H2B and histone chaperones cooperating in replication-independent nucleosome assembly (*Wu et al., 2017*).

Ubiquitylation of Kap-α may also be important for its ability to promote nuclear import of a key maternal protein(s) involved in protamine-histone exchange (or for a nuclear non-transport function of Kap-α) (*Oka and Yoneda, 2018*). Our crosses suggest both Kap-α and P32 are limiting in CI embryos because transgenic expression of either suppresses *Wolbachia*-induced incompatibility. In regard to the above model, CidB deubiquitylation of ubiquitin-modified histones, histone chaperones and/or Kap-α would be envisioned to impair histone deposition (but not protamine removal [*Landmann et al., 2009*]). Overexpressed Kap-α might enhance import of histone chaperones or ubiquitylation factors to overcome the activity of CidB. Similarly, overexpression of histone chaperones such as P32 could enhance nucleosome assembly. Determination of exactly how the proteins we have identified contribute mechanistically to CI is an important goal for future studies.

The fact that the antidote, CidA, contributes to both CI induction and rescue is seemingly at odds with its designation as an antidote. However, this dual functionality is characteristic of toxin-antidote (TA) operons (*Yamaguchi et al., 2011*). Our previously described model envisioned co-translation of CidA and CidB followed by CidA-B protein complex formation, possibly after passage through a type IV secretion system into the host cytoplasm (*Beckmann et al., 2019a*; *Beckmann and Fallon, 2013*). We postulated that CidA antidote functionality has a dual

purpose. One function is to prevent premature toxicity of CidB during spermiogenesis. CidA may even promote localization of the toxin into sperm. Rapid degradation of antidote, also characteristic of TA operons, in the egg would activate the relatively stable CidB toxin if no fresh CidA is provided by egg-resident *Wolbachia*.

To reiterate, induction of CI could proceed by multiple mechanisms based on the data in hand. The simplest model is that CidB directly deubiquitylates a single key target, possibly Kap-α2. In this model ubiquitylated Kap-α2 is crucial for delivery of some key factor, perhaps P32, Nap1, or histones to the male pronucleus. Alternatively, CidB may have multiple direct targets. It might deubiquitylate many of the proteins found in *Table 1a*, for instance, and CI results from the accumulated defects caused by these changes. A more indirect model would posit that CidB binds Kap-α2 as a way into the nucleus where its relevant substrates localize. Localization studies will be crucial for determining the precise mechanisms.

Based on our fly protein interactome data (*Figure 4*), we view "rescue" as an exclusion mechanism. In this model, maternal CidA, short-lived but abundantly expressed and provided by *Wolbachia* in infected eggs, associates with the more metabolically stable CidB and prevents the deubiquitylase from binding its relevant target(s). Such binding could also cause changes in CidB localization and/or changes in its substrate preferences.

In general, Nuclear transport as a target of CI is tantalizing because it suggests divergent selfish reproductive manipulators converge on related embryonic processes. Segregation Distorter (SD) was also linked to nuclear import disruption (*Merrill, 1999*; *Larracuente and Presgraves, 2012*). SD is a meiotic driver in natural *D. melanogaster* populations involving two autosomal loci. The *Sd* driver locus encodes a truncated but catalytically active RanGAP (nuclear transport regulator) that mislocalizes to the nucleus (*Kusano et al., 2002*), and the responder (*Rsp*) locus is a large block of satellite DNA. During spermiogenesis, Sd-RanGAP alters the histone-to-protamine transition, culling drive-sensitive spermatids. Phylogenomic analysis of karyopherins in *Drosophila* also suggested frequent gain and loss of Kap-α genes, consistent with selection targeting nuclear transport for host protection against genetic conflicts (*Phadnis et al., 2012*). Independently, a *Drosophila* testes-specific X-linked Kap-α gene was found to be duplicated and overexpressed in response to a sex-ratio driver (SR) that selectively blocks maturation of Y chromosome-bearing sperm (*Pieper et al., 2018*). Hence, the molecular features of SD and SR show remarkable parallels with the processes we have linked to CI, particularly nuclear transport (Kap-α, Moleskin) and the protamine-to-histone transition (P32 and Nap1).

## Host suppression of CI

Host suppression of reproductive parasitism has been documented in multiple *Wolbachia* systems involving CI (*Cooper et al., 2017*) and male killing (*Hornett et al., 2006*; *Reynolds et al., 2019*). Theory predicts that CI will progressively evolve to weaker incompatibilities (*Turelli, 1994*; *Prout, 1994*). However specific suppressor gene loci have never been identified. Genetic suppressors of CI are important for two reasons. First, they provide hints toward pathways targeted by CI. Secondly, they might co-evolve as resistance factors to CI. Importantly, suppression of CI in vectors will reduce the effectiveness of global mosquito control efforts harnessing *Wolbachia* and CI. We note that Kap-α and P32 were both robust dosage suppressors of transgenic and natural CI (*Figure 5*) and both are maternally deposited (*Emelyanov et al., 2014*; *Emelyanov and Fyodorov, 2016*; *Mason et al., 2002*). Therefore, these proteins could well be important factors in the evolution of host resistance to *Wolbachia*-induced CI.

# Materials and methods

## Nucleic acid sources and construct preparation

Yeast genomic DNA was purified by lysing cells by glass bead disruption, followed by phenol/chloroform extraction and ethanol precipitation (*Hoffman and Winston, 1987*). *Wolbachia* DNA was purified by homogenizing 10 whole infected insects in lysis buffer and recovering DNA with organic extraction following referenced protocols (*Beckmann et al., 2017*; *Beckmann and Fallon, 2012*). *Drosophila melanogaster* and *D. simulans* lines infected with *w*Mel, *w*Ri, and *w*Ha were used as PCR template sources. In some cases, genes from *w*No and *w*Mel were subcloned from synthesized

constructs (Genscript). Genomic wStr DNA was a gift from Ann Fallon and was derived from infected cell cultures. PCR amplicons were produced with primers listed in *Supplementary file 1i*. High fidelity Phusion polymerase (New England Biolabs) was used to amplify DNA, which was then restriction enzyme digested, gel-purified and ligated into various plasmid vectors (*Supplementary file 1j*). Plasmids were sequenced and confirmed at the Yale Keck Foundation DNA sequencing facility. Point mutations were introduced by QuikChange site-directed mutagenesis (Stratagene). Other modifications such as truncations or tag additions/swaps were created by site-directed ligase-independent mutagenesis (SLIM) (*Chiu et al., 2004*).

## Yeast methods

Yeast strain backgrounds used were W303-1A and BY4741. BY4741 was discontinued after *Figure 1a* because W303-1A exhibited stronger sensitivity to CI factors. All other serial dilution and Western blotting data used W303-1A except *Figure 3d* which was BY4741. Yeast were transformed with plasmids by standard methods (*Gietz and Schiestl, 2007*). In general, *cifB* gene toxins were expressed from low-copy CEN vectors under control of the *GAL1* promoter. When testing co-expression with *cifA* genes or suppressors we placed these latter genes in high-copy 2-micron plasmids, with the *cifA* genes also under control of the *GAL1* promoter. Suppressors were always expressed under endogenous promoters. For specific plasmid descriptions see the construct database, *Supplementary file 1j*. Five-fold serial dilutions and plating of yeast cultures were described previously (*Beckmann et al., 2017*). SDS-PAGE and Western blotting analysis of yeast protein extracts was performed precisely as detailed in our prior work (*Beckmann et al., 2017*). All serial dilution and Western blot data are representative of at least three biological triplicate experiments.

## Yeast suppressor screens

Plasmids from a yeast high-copy ordered genomic library were purified from *E. coli* (Qiagen) and stored at −80˚C (*Jones et al., 2008*). $^{His6}CidB^{wPip}$ was cloned into the pRS416GAL1 plasmid using BamHI-5' and XhoI-3' restriction sites. This low-copy CEN plasmid has a galactose-inducible promoter and a *URA3* cassette. Suppressor screens were performed in the BY4741 yeast background. BY4741 [pRS416GAL1-$^{His6}CidB^{wPip}$] yeast were streaked out on synthetic defined (SD) glucose medium lacking uracil (SD-ura). An overnight liquid starter culture (5–10 ml) was inoculated in SD-ura medium at 30˚C. The following day, cultures were diluted and allowed to grow to $OD_{600}$ 0.8 in SD-ura at 30˚C. Cells were pelleted by centrifugation, washed in sterile water and transformed using lithium acetate transformation with 17 sublibrary plasmid minipreps (*Gietz and Schiestl, 2007*).

Transformed cells were plated directly on selective medium (synthetic defined galactose medium lacking uracil and leucine) at a range of temperatures. This selects for the *URA3* and *LEU2* markers in the toxin-expressing and library plasmids, respectively, and galactose induces expression of $^{His6}$-$CidB^{wPip}$ which kills yeast unless they carry a suppressor. More than five iterative screens were performed under varying conditions, including temperatures of 37, 36.5, 34, and 33˚C. We also tested variant strategies of plating transformants on the selective media. If we first plated cells on glucose media, allowed the transformants to grow into colonies, and then replica-plated onto galactose media, it yielded high background and more false positives. These methods and the plasmids identified are summarized in *Supplementary file 1a*.

After plating on selective media, colonies were allowed to grow for 3–7 days, which helped colony sizes diverge according to suppressive capability. Potential suppressors were then re-streaked under the same selective conditions. Yeast colonies were then inoculated into 2 ml of yeast peptone dextrose (YPD) liquid medium and allowed to grow for two days to high density. Cultures were used for 'smash-n-grab' plasmid recovery (*Hoffman and Winston, 1987*). Recovered DNA was electroporated into electrocompetent Top10F' *E. coli* and plated on LB plates containing kanamycin. The ends of the recovered plasmid inserts were sequenced with primers JFB 146 and 147 (*Supplementary file 1i*). Sequencing data were cross referenced with the *S. cerevisiae* genome using NCBI BLAST (*Figure 2a*, *Supplementary file 1a*). Identified plasmids were re-transformed back into yeast and tested by serial dilution to confirm suppression (*Figure 2b*). To identify individual suppressor genes from these library plasmids, we sub-cloned each gene individually into the library vector pGP564. These clones were then transformed into yeast and tested by serial dilution (*Figure 2c*).

## Recombinant protein expression and substrate trapping interactomes

All recombinant protein expression constructs and isolation protocols were described previously (*Beckmann et al., 2017*). We used similar protocols with some minor modifications listed here. Recombinant proteins were expressed in BL21-AI (ThermoFisher). N-terminally His6-tagged proteins expressed from an arabinose inducible promoter in the plasmid pBAD (ThermoFisher) were purified by affinity chromatography using HisPur cobalt resin (Qiagen). Three constructs were used to produce three interactomes in (*Figure 4*): $^{His6}$CidA$^{wPip}$, $^{His6}$CidB*$^{wPip}$ (C1025A), and $^{FLAG}$CidA$^{wPip}$. *Drosophila melanogaster* lysates (male and female adults) were run over the column to enrich for insect proteins capable of binding the recombinant proteins. For detailed expression, purification, and pull-down protocols, see below. In-solution LC-MS/MS analysis was performed at the Yale Keck Foundation in close association with the authors (for details see below).

## Transgenic *Drosophila* and transgenic CI crosses

DNA for the *cidA-T2A-cidB*$^{wMel}$ operon (*Beckmann et al., 2017*), in addition to D.m.Kap-$\alpha$1 were codon optimized for *Drosophila* and ordered from Genscript. Some constructs were purchased from Genscript *Drosophila* cDNA libraries. Transgenes were sub-cloned from the pUC57 vector into pUASp-attB (*Rørth, 1998*; *Takeo et al., 2012*). This vector appends the K10 3' UTR, which is known to localize transcripts to the *Drosophila* oocyte (*Serano and Cohen, 1995*). Final constructs were either fully sequenced or sequenced on ends and verified by restriction enzyme digests. Cloning and construct specifics are in *Supplementary file 1j*. BestGene Inc was contracted for embryo microinjection of *D. melanogaster* #9744 (attP site on chromosome three) and ΦC31 integrase-mediated transgene insertion. We verified that all fly lines were free of *Wolbachia* using PCR and primers recognizing a conserved region in the *Wolbachia* VirD4 gene (*Figure 5—figure supplement 1*). As a positive control, we amplified the *D. melanogaster* histone H3 gene. Crossing of *cidA-T2A-cidB*$^{wMel}$ operon-transformed male flies with females from strain #4442 carrying the nanos-Gal4-tubulin 3' untranslated region (NGT) driver induced CI (*Figure 5a*). This served as a phenotypic confirmation of transgene expression and accords with previous results (*Beckmann et al., 2017*; *LePage et al., 2017*).

Flies were maintained on a standard diet, and temperature was stringently controlled as outlined in *Figure 5*. For CI analysis, $F_0$ crosses were initiated by crossing homozygous Gal4 driver females to homozygous UAS-transgene males. $F_0$ crosses were kept at the temperatures indicated in *Figure 5* to control for any temperature-dependent maternal effects. Temperature was only temporarily lowered to 18°C for overnight virgin collection. $F_1$ flies, which were heterozygous for both the NGT driver and the Gal4-UAS-transgene, were aged 3–4 days at restrictive temperature and crossed one to one, male and female, in arenas with apple juice plates and yeast paste. After 12 hr, we discarded the original apple juice plate and allowed flies to oviposit for 24 hr before removing the plate. Eggs were given 36 hr to hatch while being incubated at the respective temperatures. Hatch rates were evaluated by microscopy and by counting hatched and unhatched egg totals. One-way ANOVA with multiple comparison was performed using Graphpad Prism seven with outliers removed by the ROUT method. Flies used in this study were *white* Canton-S ($^w$CS; WT); *nanos-Gal4-tubulin*, #4442; MTD-Gal4, #31777, which has multiple GAL4 inserts on all three large chromosomes, including *nanos-Gal4*, *nanos-Gal4:VP16*, and *otu-Gal4* and is infected with *Wolbachia*; and UASp-Kap-$\alpha$2, #25400 (*Mason et al., 2002*). Fly lines were created by us, obtained from the Bloomington Stock Center, or were gifts.

## Rationale for dual CidA and CidB transgenic CI expression

The *cidB*$^{wPip}$ gene alone was not successfully inserted into flies in >600 embryo microinjections (*Beckmann et al., 2017*). Our interpretation of this observation was that CidB$^{wPip}$ might be toxic by itself and was killing the injected flies. In order to build a transgenic fly that expressed CidB$^{wPip}$, we reasoned that co-expressing it with the upstream CidA$^{wPip}$ protein might alleviate this toxicity. We built the fusion ORF cidA$^{wPip}$ –T2A–cidB$^{wPip}$, where T2A encodes a viral peptide that causes ribosomal skipping and translation of the upstream and downstream polypeptides at roughly 1:1 stoichiometry. This worked. Importantly, this system mimics what would occur in a normal *Wolbachia* infection and other natural toxin-antidote systems in which both proteins are expressed simultaneously. Toxicity in most known toxin-antidote systems occurs only after rapid degradation of the

antidote in cells that no longer synthesize it. This activates the toxin. We hypothesized that the reason CI was induced was because there was a similar rapid degradation of CidA antidote in the fertilized egg, although this remains to be shown experimentally (*Beckmann et al., 2019a*; *Beckmann et al., 2017*).

In *LePage et al. (2017)* CI factors from another *Wolbachia* strain were used, namely CidA-CidB$^{w-Mel}$. The CI system of *w*Mel has traits making it different from the *w*Pip system described above. It is a much weaker CI inducer. In that publication, a transgenic fly line with cidB$^{wMel}$ alone was generated and did not induce CI. Only when combined with an insertion of cidA$^{wMel}$ on another chromosome was CI induced. The requirement for both proteins for CI in these transgenic models could either be that both are needed for interference with embryonic nuclear division or, as suggested above, that CidA promotes CI indirectly by preventing premature toxicity of CidB in the male and/or promoting CidB packaging into sperm. Because this dual expression system was able to induce transgenic CI, we replicated it with the *w*Mel operon in this study except via a T2A peptide mechanism.

## Detailed protein expression, purification, and pull-down analysis

Bacterial starter cultures were grown overnight and used to inoculate 4 liters of Luria Broth (LB) plus ampicillin. An additional 4-liter culture of BL21-AI cells was always grown in tandem to serve as an internal mock negative control with which to rule out non-specific interactions from copurification analyses; the mock control and experimental samples were treated equivalently. Cultures were grown at 37°C with vigorous shaking to an optical density at 600 nm of 0.5 and induced with 0.02% arabinose. Immediately after induction, we shifted the cultures to 18°C and incubated overnight. The following morning cells were pelleted by centrifugation and resuspended in 15 ml of 50 mM Tris pH 8.0, 250 mM NaCl, 10% glycerol, 2 mM β-mercaptoethanol. We added 100 µl of 100 mM PMSF in isopropanol and one cOmplete, Mini, EDTA-free Protease Inhibitor Cocktail tablet (Roche).

Cells were lysed by incubation with a pinch of chicken egg-white lysozyme on ice for 30 min followed by two passes through a French-press. Lysate was centrifuged for 45 min at 30,000 x *g* (Beckman Coulter Type 50.2 TI). Following centrifugation, supernatant was decanted in a beaker on ice with a stir bar. In order to precipitate and remove DNA, we added 5 M NaCl while stirring to a final concentration of 1 M and poly(ethyleneimine) (PEI) from a stock of 10% PEI in 10% HCl to a final concentration ~0.3–0.5%. Fresh PEI solution was used to ensure efficient DNA precipitation. DNA was precipitated after 5 min stirring on ice and pelleted by centrifugation at 4700 x *g* for 15 min in a Thermo Sorvall Lynx 600 F9−6 × 1000 LEX centrifuge. Supernatant was transferred to a new tube and proteins gently precipitated by adding 0.436 g/ml ammonium sulfate on ice while stirring for 15 min. Precipitated proteins were then pelleted at 30,000 x *g* for 30 min and the supernatant removed.

Protein pellets were resuspended in wash buffer (50 mM sodium phosphate pH 8.0, 300 mM NaCl, 0.01% Tween-20, 5 mM β-mercaptoethanol, 10 mM imidazole) and run over a 10 ml disposable chromatography column containing 1 ml of fresh HisPur cobalt resin at 4°C to bind recombinant His6-tagged proteins. The columns were washed with 20 column volumes (20 ml) of wash buffer. A peristaltic pump was used to aid column flow. *Drosophila* lysates made from male and female adults were then run through the column. To prepare *Drosophila* lysates, 10 ml of fly bodies were collected and stored in −80°C. Bodies were ground to powder in liquid nitrogen by mortar and pestle and 25 ml of pull-down buffer (3.25 mM Sodium-phosphate, pH 7.4, 70 mM NaCl, 0.01% Tween-20) was added. Fly cuticle and insoluble material were pelleted by centrifugation at 4700 x g for 15 min in a Thermo Sorvall Lynx 600 F9−6 × 1000 LEX centrifuge and the supernatant was passed through a 5-micron filter (Amicon), loaded on the HisPur-recombinant protein column, and allowed to pass through the column by gravity for 1 hr. The column was then washed with 50 column volumes (50 ml) of wash buffer. Proteins were eluted with 5 ml of elution buffer (300 mM imidazole, 50 mM Sodium-phosphate pH 8.0, 300 mM NaCl, 0.01% Tween-20). Eluates were concentrated to 250 µl in an Amicon 3000 molecular weight cutoff centrifugal filter. Concentrated eluates were then subjected to in-solution proteome analysis as described below. In the case of the bound $^{His6}$CidB*$^{wPip}$/$^{FLAG}$CidA$^{wPip}$ interactome, $^{FLAG}$CidA$^{wPip}$ was added as a bacterial extract, followed by an additional wash with 20 column volumes wash buffer, and then immediate addition of the fly lysate.

## Protein digestion

Proteins were precipitated from the eluates with acetone using established protocols. Protein pellets were dissolved and denatured in 8 M urea, 0.4 M ammonium bicarbonate, pH 8. The proteins were reduced by the addition of 1/10 vol of 45 mM dithiothreitol (Pierce Thermo Scientific #20290) and incubation at 37°C for 30 min, then alkylated with the addition of 1/10 vol of 100 mM iodoacetamide (Sigma-Aldrich #I1149) with incubation in the dark at room temperature for 30 min. The urea concentration was adjusted to 2 M by the addition of water prior to enzymatic digestion at 37°C with trypsin (Promega Seq. Grade Mod. Trypsin, # V5113) for 16 hr. Protease:protein ratios were estimated at 1:50. Samples were acidified by the addition of 1/40 vol of 20% trifluoroacetic acid, then desalted using C18 MacroSpin columns (The Nest Group, #SMM SS18V) following the manufacturer's directions. Peptides were eluted with 0.1% TFA, 80% acetonitrile. Eluted peptides were dried in a Speedvac and dissolved in MS loading buffer (2% aceotonitrile, 0.2% trifluoroacetic acid). Protein concentrations were determined using a Thermo Scientific Nanodrop 2000 UV-Vis Spectrophotometer. Each sample was then further diluted with MS loading buffer to 0.08 µg/µl, with 0.4 µg (5 µl) injected for LC-MS/MS analysis.

## LC-MS/MS

LC-MS/MS analysis was performed on a Thermo Scientific Orbitrap Fusion equipped with a Waters nanoAcquity UPLC system (Yale Keck Center) utilizing a binary solvent system (Buffer A: 100% water, 0.1% formic acid; Buffer B: 100% acetonitrile, 0.1% formic acid). Trapping was performed at 5 µl/min, 97% Buffer A for 3 min using a Waters Symmetry C18 180 µm x 20 mm trap column. Peptides were separated using an ACQUITY UPLC PST (BEH) C18 nanoACQUITY Column 1.7 µm, 75 µm x 250 mm (37°C) and eluted at 300 nl/min with the following gradient: 3% buffer B at initial conditions; 5% B at 5 min; 20% B at 90 min; 35% B at 125 min; 97% B at 130 min; 97% B at 135 min; and return to initial conditions at 136–150 min. Mass spectra were acquired in the Orbitrap in profile mode over the 300–1,500 m/z range using quadrapole isolation, one microscan, 120,000 resolution, AGC target of 4E5, and a maximum injection time of 60 ms. MS/MS data were collected in top speed mode with a 3 s cycle time on species with an intensity threshold of 5E4, charge states 2–8, peptide monoisotopic precursor selection preferred. Dynamic exclusion was set to 30 s. Data-dependent MS/MS were acquired in the Orbitrap in centroid mode using quadropole isolation (window 1.6 m/z), HCD activation with a collision energy of 28%, one microscan, 60,000 resolution, AGC target of 1E5, maximum injection time of 110 ms.

## Peptide identification

Data were analyzed using Proteome Discoverer software v2.2 (Thermo Scientific). Data searching was performed using the Mascot algorithm (version 2.6.1) (Matrix Science) against a custom database containing protein sequences for CidA and CidB* as well as proteomes for *Escherichia coli*, *Saccharomyces cerevisiae*, *Wolbachia pipientis*, and *Drosophila melanogaster* proteomes (35,536 sequences total). The search parameters included tryptic digestion with up to two missed cleavages, 10 ppm precursor mass tolerance and 0.02 Da fragment mass tolerance, and variable (dynamic) modifications of methionine by oxidation and carbamidomethylated cysteine. Normal and decoy database searches were run, with the confidence level set to 95% (p<0.05). Scaffold (version Scaffold_4.8.9, Proteome Software Inc, Portland, OR) was used to validate MS/MS-based peptide and protein identifications. Peptide identifications were accepted if they could be established at greater than 95.0% probability by the Scaffold Local FDR algorithm. Protein identifications were accepted if they could be established at greater than 99.0% probability and contained at least two identified peptides.

## Interactome data analysis

Samples were each run in technical duplicate with three independent biological replicate interactome pulldowns constituting a complete interactome dataset. Proteomic datasets were viewed in Scaffold Proteome Software and the raw datasets of identified peptide spectral matches were transitioned into Microsoft Excel. Protein 'enrichment' was measured in comparison to an internal mock control lacking recombinant protein. We ranked protein hits based on normalization of the

frequency of detecting their peptide spectra in the experimental pulldown compared to the mock control. The peptide frequency (*F*) for any protein hit was calculated by the formula:

$$F = \frac{(C - M)}{C}$$

where *C* equals the total number of peptide spectral matches for any protein X detected in the CidB* sample (i.e., recombinant protein sample) and *M* is equal to the total peptide spectral matches for the same protein X detected in the mock (negative control) sample. By this calculation, protein hits with spectra uniquely present in a CidB* pull-down and also completely absent in the control have a perfect value of 1. If there is no difference in spectra and no *enrichment* the value will equal 0. Thus, proteins *enriched* can be ranked on a scale of 0 - 1 and anything not enriched will be less than or equal to zero. Proteins enriched in at least two of three replicates were compiled in Excel (*Supplementary file 1b, d and f*). We then iteratively subjected these lists to peptide spectral analysis. In this order, we culled the lists to proteins identified as *enriched* ($F \geq 0$) in all three biological replicates for each interactome (*Supplementary files 1c, e, g*). Next, we classified these hits based on predicted protein functional categories (*Figure 4—figure supplements 1*, *2,* and *3*). Then we a) eliminated hits where averages covered up inconsistencies in the technical replicates; such hits may have had multiple peptides in one technical replicate, but zero in another; b) removed hits with standard deviations in their *F*-scores greater than that of our positive control, ubiquitin; c) removed ribosomal subunits (though these hits are still visible in *Supplementary file 1c, e and g*); and d) manually inspected the tandem spectra verifying that they all contained at least three consecutive ions, ie., b5, b6, b7 and that all peaks above background were assigned to the peptide. This process produced *final* interactomes (*Table 1*). Proteins were not reported in any table if the combination of technical replicate enrichment was not $\geq 2$. Under these stringent reporting conditions, the false discovery rate (FDR) of the interactomes is zero. P-values were calculated (comparing peptide spectral matches in *C* to *M*) by a two sample T-test assuming unequal variances of the total replicates.

## Acknowledgements

We thank Drs. Michael Turelli for *w*Ha infected *Drosophila simulans*; Ann Fallon for *w*Str genomic DNA; Jean Kanyo (Yale Keck Center) for in-solution proteome analysis; and Brandon Cooper, Sylvain Charlat, Jason Berk and Chris Hickey for constructive criticism. Funding was supported by NIH grants (GM046904 and GM053756), NIH S10 (SIG) OD018034 awarded to the Mass Spectrometry (MS) and Proteomics Resource of the W.M. Keck Foundation Biotechnology Resource Laboratory at Yale University, USDA-1015922 (JFB), Auburn University's Department of Entomology and Plant Pathology Startup Funds (JFB), and an Alabama Agricultural Experiment Station SEED grant (JFB). We thank reviewers and eLife for respectful innovative peer review.

## Additional information

### Funding

| Funder | Grant reference number | Author |
| --- | --- | --- |
| USDA | 1015922 | John Frederick Beckmann |
| NIH | GM046904 | Mark Hochstrasser |
| NIH | GM053756 | Mark Hochstrasser |
| Alabama Agricultural Experiment Station | SEED Grant | John Frederick Beckmann |

The funders had no role in study design, data collection and interpretation, or the decision to submit the work for publication.

### Author contributions

John Frederick Beckmann, Conceptualization, Resources, Data curation, Formal analysis, Supervision, Funding acquisition, Validation, Investigation, Visualization, Methodology, Writing—original

draft, Project administration, Writing—review and editing; Gagan Deep Sharma, Luis Mendez, Data curation, Investigation; Hongli Chen, Conceptualization, Data curation, Investigation, Methodology, Writing—review and editing; Mark Hochstrasser, Conceptualization, Data curation, Formal analysis, Supervision, Funding acquisition, Methodology, Project administration, Writing—review and editing

#### Author ORCIDs
John Frederick Beckmann (iD) https://orcid.org/0000-0001-7398-9647
Gagan Deep Sharma (iD) https://orcid.org/0000-0002-7865-1888

#### Decision letter and Author response
Decision letter https://doi.org/10.7554/eLife.50026.sa1
Author response https://doi.org/10.7554/eLife.50026.sa2

## Additional files

#### Supplementary files
• Supplementary file 1. Supplementary data within Microsoft excel spreadsheets.a. Plasmids recovered from iterations of a CidB high copy suppression screen. b. CidB* interactome raw data c. Raw data for biological triplicate CidB* enriched hits d. CidB* + CidA interactome raw data e. Raw data for biological triplicate CidB* + CidA enriched hits f. CidA interactome raw data g. Raw data for biological triplicate CidA enriched hits h. Fly Crossing Hatch-Rate Data i. PCR primer database j. Construct database.

• Transparent reporting form

#### Data availability
All data generated or analyzed during this study are included in the manuscript and supporting files.

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
