## [Decision Letter]

**Acceptance summary:**

The ability of intracellular parasites like *Wolbachia* to induce cytoplasmic incompatibility (CI) - in which the presence of the bacteria in one or both germ cells leads to the production of inviable gametes - has been studied for decades as a window into the complex interactions between symbionts and their hosts. These studies have become more urgent in recent years as such systems are increasingly being deployed in the field to control populations of disease vectors, especially mosquitos. The authors of this work previously identified a *Wolbachia* toxin CidB that induces CI and a corresponding antitoxin CidA that suppresses it. This paper addresses the question of what processes are targeted by CidB, using a combination of genetic screens and biochemistry to identify a nuclear import factor and a factor involved in the packaging of sperm DNA as likely targets, which they confirm with follow up functional studies. The reviewers found the experimental design and execution to be of uniformly high quality, and the data convincing. Given the abundance of manuscripts on the phenomenology of CI, mechanistic studies such as the one presented here are essential, and certain to be of immediate interest to both the symbiosis and vector control communities.

**Decision letter after peer review:**

Thank you for submitting your article "*Wolbachia* Cytoplasmic Incompatibility Enzyme CidB Targets Nuclear Import and Protamine-Histone Exchange Factors" for consideration by *eLife*. Your article has been reviewed by two peer reviewers, and the evaluation has been overseen by Michael Eisen as the Senior and Reviewing Editor. The following individual involved in review of your submission has agreed to reveal their identity: William Sullivan (Reviewer #1).

The reviewers have discussed the reviews with one another and the Reviewing Editor has drafted this decision to help you prepare a revised submission.

The manuscript by Beckmann et al. follows up previous studies that identified the genes responsible for *Wolbachia*-induced CI and rescue. Over the past decade the relevance of these studies has greatly increased because of mass releases of *Wolbachia*-infected mosquitos to prevent insect borne diseases. This strategy relies on *Wolbachia*-induced CI, thus elucidating the underlying mechanisms has immediate relevance for vector control.

Previously the authors identified *Wolbachia*-expressed CidB, a deubiquitylase (DUB), as causing CI and *Wolbachia* Cid A as the maternal rescuing element. Using a yeast expression system and binding assays they provided strong experimental support for CidB and CidA operating as a toxin antitoxin respectively.

Here, using parallel screens in yeast and flies, they uncover candidate targets of the *Wolbachia* toxin, CidB. One target that emerged from both screens is the nuclear import factor, karyopherin. The authors show that overexpression of this factor in *Drosophila* embryos suppresses CidB toxicity (embryo viability increases). Similarly, overexpression of P32, a factor uncovered only by their CidB protein interaction search using *Drosophila* adult lysate, also suppresses CidB toxicity in vivo. These discoveries identify not only key molecules and mechanisms controlling CI (with implications for insect vector and pest biology), but also possibly a new maternal factor that processes the paternal genome in the early embryo (in the absence of *Wolbachia* infection). The experimental design and quality of the data is excellent and with a few exceptions the writing is clear and concise.

Given the overabundance of manuscripts on the phenomenology of CI, mechanistic studies such as these are essential sorely needed contributions and certain to be of immediate interest to the symbiosis and vector control communities. However the reviewers raised a number of issues that must be addressed prior to acceptance:

Essential revisions:

1) A major strength of the paper is the convergence of a candidate CidB target from yeast and *Drosophila*-based assays. The toggling between different *Wolbachia* strains throughout the paper suggests common mechanisms of CI. However, the authors also emphasize the CidB toxin requires cognate CidA for potent suppression (e.g., Figure 1B). The authors need to address the seemingly paradoxical strain-specificity of the *Wolbachia* Cif protein interactions and the putative kingdom-wide conservation of CidB targets. The results are especially surprising given the expectation that these host targets evolve rapidly to evade *Wolbachia* antagonism.

2) The data presented in Figure 5 (demonstrating suppression of *Drosophila* CI through maternal expression of genes identified in the screen) needs clarification:

a) Based on Materials and methods, each data point is an egg lay from 1X1 mating and the% hatch. To evaluate this an N (total number eggs) needs to be included for each 1X1 mating). For each cross, it would also be useful to have the sum of total # hatched/total # of eggs. Also the data points with many eggs and no hatch are likely due to laying of unfertilized eggs from unmated females – these should be excluded from the analysis.

b) A clearer explanation needs to be given concerning the rescuing effect of the GFP alone compared to the karyopherin and accounting for karyopherin induced lethality.

c) The fact that a diverse array of transgenes result in rescue is intriguing suggesting may be also occur through non-specific effects (slowing develop etc). This would also fit with the studies presented demonstrating different yeast backgrounds exhibit different toxicities (Figure 1).

d) For those not intimately familiar with recent CI literature, it would be helpful to include an explanation of why expression of both CidB/CidA is being driven in the male (Given the toxin (CidB)/ Antitoxin (CidA) model presented the rational for this approach is not obvious). The dual expression is also described in their Introduction with no additional explanation. Perhaps these sections could point the reader to an expanded Materials and methods section where the rational is fully explained.

3) Figure 5 strongly supports the capability of both P32 and Kap-α to suppress CI in the embryo. The relevance of P32 to CI is clear -- maternally deposited P32 processes sperm-deposited paternal chromosomes in uninfected embryos and in CI, the paternal chromosomes fail to condensed properly at the first zygotic prophase. However, Kap-α has yet to be implicated in paternal genome processing post-fertilization. Do we know if Kap-α is maternally deposited? The absence of either IF- or Western – based evidence that Kap-α is detectable prior to zygotic genome activation weakens the implied relevance of Kap-α to CI in nature. This needs to be addressed by the authors.

4) Related to above, do the authors imagine that the toxin targets different developmental events between sperm entry and the first zygotic division and that is why targets with different roles in this process emerged from the *Drosophila*-based screen? Do the authors imagine Kap-α and P32 act independently during this short developmental stage? What is the model?

5) Also related to the above, a model that reconciles previous work showing that the earliest CI-defect in *Drosophila* embryos appears at prophase with the current work that shows that, based on CidB targets, CI compromises transition to nucleosome-based chromatin on paternal DNA. Based on the discovery that CidB binds P32, we would expect instead that sperm-deposited DNA would fail to decondense after sperm entry and ultimately fail to participate at all in the first zygotic mitosis.

6) The proteomic analysis demonstrates that CidA binding to CidB* does not compromise interactions with other proteins. The (surprisingly?) expanded set of interactors under this treatment warrants further comment.

7) The Introduction could benefit tremendously from some restructuring and addition of important background information. For example, the second paragraph of the Introduction asserts that 'CidA and CidB proteins precisely mimic naturel CI.' However, the reader does not yet know what 'natural CI' looks like. Indeed, the word "embryo" has not yet appeared in the Introduction. The fifth paragraph of the Introduction could be moved up to help explain natural CI. In addition, CidB*^w^*^Pip^ (Introduction, second paragraph) is not explicitly defined as derived from *Culex pipiens*. The third paragraph of the Introduction could also benefit from additional information about how CidA is 'inferred from bi-directional crosses." Generally, the Introduction appears to be written in "short format" and so fails to completely introduce the system to the uninitiated reader.

8) Where are the data for the high copy suppressor plasmids encoding the other sub-cloned genes that did not exhibit suppressor activity? (Subsection “Yeast Dosage Suppressors of cidB Toxicity, last paragraph; Figure 2). These data should be included as supplement.

9) Why were only Kap-α and P32 used in the final CI suppression assay? Do the other candidates not suppress CI?

10) The Discussion does not directly address many of key findings of the manuscript and how it relates to previous work on CI. For example, the Abstract states "CidB targets nuclear-protein import and protamine-histone exchange and that CidA rescues embryos by restricting CidB access to its target" yet no mention is made of this idea in the Discussion. There are other examples as well.

---

## [Author Response]

Essential revisions:1) A major strength of the paper is the convergence of a candidate CidB target from yeast and Drosophila-based assays. The toggling between different Wolbachia strains throughout the paper suggests common mechanisms of CI. However, the authors also emphasize the CidB toxin requires cognate CidA for potent suppression (e.g., Figure 1B). The authors need to address the seemingly paradoxical strain-specificity of the Wolbachia Cif protein interactions and the putative kingdom-wide conservation of CidB targets. The results are especially surprising given the expectation that these host targets evolve rapidly to evade Wolbachia antagonism.

There is no real paradox here. The specificity of “rescue” is conveyed by cognate CidA-CidB binding but toxin substrate specificity is expected to derive from the “warhead” of CidB, which is the deubiquitylase (DUB) domain. The targets of the DUB activity can be the same even across phylogenetically diverse taxa. Differential binding among various CidA and CidB variants, and therefore the possibility of rescue, is predicted to be due to sequence differences in their binding interfaces. Examination of sequence variation in CidA and CidB alleles compared to crossing type diversity among geographically separated *Culex pipiens* mosquito populations infected with *w*Pip *Wolbachia* variants is fully consistent with this view. Rescue through antidote binding to the CidB toxin does not result from direct inhibition of the DUB catalytic site by CidA; instead, it appears to be a consequence of limiting access of the toxin to its key targets in vivo. We have addressed these points in the revised Discussion section.

2) The data presented in Figure 5 (demonstrating suppression of Drosophila CI through maternal expression of genes identified in the screen) needs clarification:a) Based on Materials and methods, each data point is an egg lay from 1X1 mating and the% hatch. To evaluate this an N (total number eggs) needs to be included for each 1X1 mating).

We agree. We have included the N values on the righthand side of the figure for each cross.

For each cross, it would also be useful to have the sum of total # hatched/total # of eggs.

We agree. We have added the additional Supplementary file 1H with this data.

Also the data points with many eggs and no hatch are likely due to laying of unfertilized eggs from unmated females – these should be excluded from the analysis.

We agree; We hypothesize that unmated females lay fewer eggs than mated females. In our hands, healthy mated females lay around 30 – 90 eggs. In our datasets, we excluded egg lay data if the females lay < 5 eggs. Thus, we have done our best to eliminate possible data relics induced by unmated females through this process. We also point out that our controls do not have any unmated females (values of zero hatch-rate), so we do not feel this issue impacts our datasets.

b) A clearer explanation needs to be given concerning the rescuing effect of the GFP alone compared to the karyopherin and accounting for karyopherin induced lethality.

We have attempted to clarify this in the modified text; please see 2c response below.

c) The fact that a diverse array of transgenes result in rescue is intriguing suggesting may be also occur through non-specific effects (slowing develop etc). This would also fit with the studies presented demonstrating different yeast backgrounds exhibit different toxicities (Figure 1).

We agree that in the transgenic CI experiments, non-specific suppression was observed, although non-specific suppressive effects were typically weak (~15% in the case of GFP). Thus, we only inferred mechanistically relevant suppression in cases where the observed suppression was significantly greater than with GFP. With P32 and Kap-α2 we also saw strong suppression against natural CI (males infected with *w*Mel), and in that case non-specific suppression did not result when crossed to females expressing GFP.

The fact that GFP weakly suppressed transgenic CI but not natural *Wolbachia*-induced CI suggests that experiments utilizing the Gal4/UAS systems can produce non-specific suppression. Toxicity due to CI might occur at multiple stages. Some embryos die in the first round of zygotic nuclear division, with escapees dying at later stages (Beckmann, Ronau and Hochstrasser, 2017; Callaini et al., 1996). One possible explanation of weak non-specific suppression could be that crosses to mothers with a UAS-driven transgene alleviate a later, secondary stage of CI killing by reducing embryonic expression of the transgenic CidB toxin in older embryos. This might result from competitive binding of the Gal4 transcription factor onto the mother’s UAS insertions, limiting binding to the transgenic *UAS-cidB* gene. Because of non-specific suppression, we urge caution about conclusions with suppression of transgenic CI. Suppression data should be backed by suppression analysis against natural bacteria induced CI, as we have done for P32 and Kap-α. We added commentary on these issues to the Discussion section.

d) For those not intimately familiar with recent CI literature, it would be helpful to include an explanation of why expression of both CidB/CidA is being driven in the male (Given the toxin (CidB)/ Antitoxin (CidA) model presented the rational for this approach is not obvious). The dual expression is also described in their Introduction with no additional explanation. Perhaps these sections could point the reader to an expanded Materials and methods section where the rational is fully explained.

We agree and have modified the Introduction to include a brief discussion of these points. We also added an additional subsection “Rationale for dual CidA and CidB transgenic CI expression” to the Materials and methods.

3) Figure 5 strongly supports the capability of both P32 and Kap-α to suppress CI in the embryo. The relevance of P32 to CI is clear -- maternally deposited P32 processes sperm-deposited paternal chromosomes in uninfected embryos and in CI, the paternal chromosomes fail to condensed properly at the first zygotic prophase. However, Kap-α has yet to be implicated in paternal genome processing post-fertilization. Do we know if Kap-α is maternally deposited? The absence of either IF- or Western – based evidence that Kap-α is detectable prior to zygotic genome activation weakens the implied relevance of Kap-α to CI in nature. This needs to be addressed by the authors.

We thank the reviewer for these thoughtful comments. In our study, transgenically expressed Kap-α, and all our transgenes for that matter, should be maternally deposited because the plasmid pUASp-attb encodes a 3’ UTR from the *Drosophila* K10 gene appended to the transgene insert. This 3’UTR was specifically added because it was shown to mediate oocyte localization of the K10 mRNA transcript (Serrano and Cohen, 1995). Pernille Rorth validated this in the original study, “Gal4 in the *Drosophila* female germline,” which describes the construction and analysis of this vector. We added a note on this in the Materials and methods. We have not verified immunologically that any of the transgenically expressed proteins are deposited in the egg.

Regarding the question of whether naturally expressed Kap-α2 is maternally deposited, data suggest that this is likely the case because it is essential for female fertility and is thought to play a paralog-specific role in oogenesis. Overexpression of maternally deposited Kap-α via transgenic constructs complements female sterility induced by homozygous Kap-α2 knockouts (see Mason et al., 2002). We added a note on this in the Discussion.

4) Related to above, do the authors imagine that the toxin targets different developmental events between sperm entry and the first zygotic division and that is why targets with different roles in this process emerged from the Drosophila-based screen? Do the authors imagine Kap-α and P32 act independently during this short developmental stage? What is the model?

CidB targeting of multiple embryonic events or substrates is possible but will take a good deal of work to prove. Our data show that CidB physically interacts with Kap-α and P32, among other proteins. One can imagine a range of models to explain these interactions as well as the ability of the overexpressed proteins to suppress CidB toxicity. One possible model is that CidB cleaves ubiquitin from both Kap-α and histone chaperones such as P32 (or histones themselves). Histone H2A and H2B are well characterized ubiquitylated proteins, and yeast Kap-α (Srp1) is ubiquitylated in vivo. There are data for ubiquitin-H2B and histone chaperones cooperating in replication-independent nucleosome assembly (Wu et al., 2017). In this model, ubiquitylation of Kap-α is needed for its import function, at least for certain substrates such ubiquitin ligases or their cofactors that act on ubiquitylated histones/chaperones. CidB itself might also be imported into the male pronucleus via Kap-α. In the nucleus, CidB activity against ubiquitylated histones or histone chaperones may further impair histone deposition (but not protamine removal). Overexpressed Kap-α, in this view, would enhance import of histone ubiquitylation factors to overcome the activity of the CidB against these proteins. In parallel, overexpression of histone chaperones such as P32 would enhance nucleosome assembly.

The above considerations are given in the extended Discussion.

5) Also related to the above, a model that reconciles previous work showing that the earliest CI-defect in Drosophila embryos appears at prophase with the current work that shows that, based on CidB targets, CI compromises transition to nucleosome-based chromatin on paternal DNA. Based on the discovery that CidB binds P32, we would expect instead that sperm-deposited DNA would fail to decondense after sperm entry and ultimately fail to participate at all in the first zygotic mitosis.

We appreciate the reviewers pointing out this subtlety. Landmann et al., 2009, who studied CI embryo cytology, reported normal removal of protamines, normal decondensation of paternal chromatin in CI embryos, but impaired deposition of maternal histone H3.3. This yielded a failure of paternal chromosomes to re-condense in a timely way as the embryos entered mitosis. We do not disagree with the Landmann et al. study and indeed have found it crucial to informing the analysis of our data. The simplest resolution to the findings that CidB binds P32 and that P32 in high dosage suppresses CI is CidB activity interferes specifically with nucleosome assembly but not protamine removal. Perhaps ubiquitylation is only required for the latter functionality of P32. We add a note on this in the Discussion. At this point, we do not definitively conclude that P32 is an actual substrate of CidB deubiquitylation. It might be equally likely, as in the model mentioned above, that CidB uses an interaction with P32 to bring it to the actual substrate(s), such as histone H2A or H2B. Our identification of the P32-CidB interaction now allows us to probe these hypotheses in detail.

6) The proteomic analysis demonstrates that CidA binding to CidB* does not compromise interactions with other proteins.

This statement is actually contradicted by our proteomic data (Figure 4) showing that CidA profoundly alters CidB*-protein binding. For example, with only CidB* on the column, it binds P32 and Kap-α2; but when complexed with CidA, it does not bind either of these targets. These findings lead directly to our inference that the rescue mechanism of CI by CidA is that it renders CidB incapable of associating with its relevant host targets.

The (surprisingly?) expanded set of interactors under this treatment warrants further comment.

There are definitely additional interactors that might be relevant to the CI mechanism. Our initial aim has been to focus on hits that either came from orthogonal screens (Kap- α) or were internally reinforced by other hits (both P32 and Nap1 are protamine-histone exchange factors, but P32 was the stronger hit). Nevertheless, we now discuss additional interactors found in proteomic data, specifically the fact that multiple AP-3 complex subunits were identified (Table 1B) in the Discussion.

7) The Introduction could benefit tremendously from some restructuring and addition of important background information. For example, the second paragraph of the Introduction asserts that 'CidA and CidB proteins precisely mimic naturel CI.' However, the reader does not yet know what 'natural CI' looks like. Indeed, the word "embryo" has not yet appeared in the Introduction. The fifth paragraph of the Introduction could be moved up to help explain natural CI. In addition, CidB^wPip^ (Introduction, second paragraph) is not explicitly defined as derived from Culex pipiens. The third paragraph of the Introduction could also benefit from additional information about how CidA is 'inferred from bi-directional crosses." Generally, the Introduction appears to be written in "short format" and so fails to completely introduce the system to the uninitiated reader.

We agree and have restructured the Introduction as suggested into a longer and more detailed form.

8) Where are the data for the high copy suppressor plasmids encoding the other sub-cloned genes that did not exhibit suppressor activity? (Subsection “Yeast Dosage Suppressors of cidB Toxicity, last paragraph; Figure 2). These data should be included as a supplement.

We have added all our screening results as Figure 2—figure supplement 2.

9) Why were only Kap-α and P32 used in the final CI suppression assay? Do the other candidates not suppress CI?

For this study we counted 53,347 *Drosophila* eggs. Due to the time, expense, and labor needed to perform CI crosses with high N, we focused our analysis on factors that we thought would be most relevant. We extensively analyzed these data with transgenic crosses and wild-type CI crosses. However, in response to the reviewers, we have now added two additional control crosses to the final suppression analysis, including crosses against Kap-α1 and cidA*^w^*^Mel^ (Figure 5D). These additional crosses add further to rigor and reproducibility of our results.

10) The Discussion does not directly address many of key findings of the manuscript and how it relates to previous work on CI. For example, the Abstract states "CidB targets nuclear-protein import and protamine-histone exchange and that CidA rescues embryos by restricting CidB access to its target" yet no mention is made of this idea in the Discussion. There are other examples as well.

We have expanded and restructured the Discussion.